**TOOLS**

# CLEM*Site*, a software for automated phenotypic screens using light microscopy and FIB-SEM

José M. Serra Lleti[1], Anna M. Steyer[1]*, Nicole L. Schieber[1]*, Beate Neumann[2], Christian Tischer[2], Volker Hilsenstein[2], Mike Holtstrom[4], David Unrau[4], Robert Kirmse[3], John M. Lucocq[5], Rainer Pepperkok[1,2], and Yannick Schwab[1]

In recent years, Focused Ion Beam Scanning Electron Microscopy (FIB-SEM) has emerged as a flexible method that enables semi-automated volume ultrastructural imaging. We present a toolset for adherent cells that enables tracking and finding cells, previously identified in light microscopy (LM), in the FIB-SEM, along with the automatic acquisition of high-resolution volume datasets. We detect the underlying grid pattern in both modalities (LM and EM), to identify common reference points. A combination of computer vision techniques enables complete automation of the workflow. This includes setting the coincidence point of both ion and electron beams, automated evaluation of the image quality and constantly tracking the sample position with the microscope's field of view reducing or even eliminating operator supervision. We show the ability to target the regions of interest in EM within 5 μm accuracy while iterating between different targets and implementing unattended data acquisition. Our results demonstrate that executing volume acquisition in multiple locations autonomously is possible in EM.

## Introduction

EM of cultured cells provides unique access to detailed subcellular architectures at a nanometer scale. Sampling strategies are essential to ensure an accurate morphometric evaluation of subcellular phenotypes. In cases where cells are homogeneous, random sampling guarantees the optimal selection of the overall population (Lucocq, 1994; Lucocq and Hacker, 2013; Gundersen and Jensen, 1987). However, different paradigms are necessary to measure subcellular morphologies in heterogeneous cell cultures (Offner et al., 1991; Altschuler and Wu, 2010). Increasing imaging throughput is one way to address heterogeneity, but EM rarely achieves sufficient regimes. Correlative light and electron microscopy (CLEM) is an efficient solution to overcome such heterogeneity in EM. It capitalizes on the power of light microscopy (LM) to screen large samples for choosing cell subpopulations of interest. By applying a selection process on the light microscopy level, analysis can be focused on specific individual cells, even if the phenotype of interest is extremely rare. Thus, various targeting strategies have been developed since the very first CLEM was performed on cultured cells (Porter et al., 1945; Porter, 1953). Individual areas of interest inside the sample can be tagged employing laser-etched frames (Colombelli et al., 2008), or cells can be seeded onto dedicated substrates that incorporate a coordinate system (Jiménez et al., 2010; Beckwith et al., 2015). In both cases, object correlation is

established using landmarks created with artificial fiducial markers that are easily identifiable in both LM and EM. Over the years, various solutions have been developed to imprint such fiducials, such as gold or ink printing (Padman et al., 2014; Prabhakar et al., 2018), laser or scalpel etching (Jiménez et al., 2010; Spiegelhalter et al., 2010), or carbon evaporation (McDonald et al., 2010).

Nowadays, commercial CLEM dishes or coverslips are routinely used for correlating fluorescence imaging of fixed or living cells with transmission EM (TEM; Stierhof et al., 1994; Polishchuk et al., 2000). Typical sample preparation for EM, i.e., by chemical fixation or high-pressure freezing, includes a resin embedding step. Upon removal of the coverslip from the resin block, the region of interest (ROI) is located using the topology of the coordinate system that marks the block surface. For TEM imaging, the block is then trimmed so the sections containing an ROI can fit onto an EM grid. Regardless of the initial dimensions of the substrate, selecting the ROI usually entails the loss of surrounding areas, preventing the analysis of multiple cells if they were distributed across the full surface of the culture dish or coverslip.

In recent years, volume scanning electron microscopy (SEM) modalities have been used for CLEM on cultured cells. Besides offering access to large volumes, both serial block-face SEM

---

[1]Cell Biology and Biophysics Unit, European Molecular Biology Laboratory, Heidelberg, Germany; [2]Advanced Light Microscopy Facility, European Molecular Biology Laboratory, Heidelberg, Germany; [3]Carl Zeiss Microscopy GmbH, Jena, Germany; [4]Fibics Incorporated, Ontario, Canada; [5]Medical and Biological Sciences, Schools of Medicine and Biology, University of St. Andrews, St. Andrews, UK.

*A.M. Steyer and N.L. Schieber contributed equally to this paper. Correspondence to Yannick Schwab: yannick.schwab@embl.de; Anna M. Steyer: steyer@embl.de.



(SBF-SEM; Titze and Genoud, 2016) and array tomography (Hayworth et al., 2015; Kislinger et al., 2020) also require block trimming before imaging and therefore suffer from the same limitations as TEM when utilized for CLEM. Focused ion beam SEM (FIB-SEM; Russell et al., 2017) however can accommodate the imaging of large specimens without the need for trimming. In particular, multiple cultured cells grown on a Petri dish or coverslip can be imaged in a CLEM workflow, even when scattered across the full surface of the substrate (Cosenza et al., 2017). Despite this capability, CLEM has been performed one cell at a time and for a limited number of cells (Narayan and Subramaniam, 2015; Cosenza et al., 2017; Fermie et al., 2018; Luckner and Wanner, 2018b), because up to now, FIB-SEM microscopes lack automation procedures to acquire multiple sites without interruption.

In this article, we introduce CLEM*Site*, a software prototype that automates serial FIB-SEM imaging of cells selected previously by fluorescence microscopy. We show that automation is not only possible but also significantly reduces the number of required manual interventions during EM imaging. In addition to the automation process, we also describe the system of landmark correlations used to find targeted cells spread over the surface sample. Our software was evaluated in two types of CLEM experiments, each experiment type was repeated twice. In the first type of experiment, for each session, we selected around 25 cells from the same dish, each cell belonging to a different phenotype. In the second experiment, the same amount of cells were selected randomly, this time with only one phenotype present in the dish. We collected a significant number of EM images from multiple cells, which allowed us to conduct morphometric analysis on different phenotypes.

## Results

### Introduction

By following the logical workflow of a CLEM experiment, CLEM*Site* was designed modularly (Fig. 1 a). The first module, CLEM*Site*-LM, is a stand-alone application to process the sets of images acquired by light and fluorescence microscopy. CLEM*Site*-LM primarily extracts stage coordinates of target cells and their associated landmarks. The second module, CLEM*Site*-EM, is divided into three components that assist with automation: *Navigator* to find and precisely navigate to the targets, *Multisite* to trigger a FIB-SEM run on each position, resulting in a stack of serial images of the corresponding ROI, and *Run Checker* to supervise operations during each acquisition. To control the FIB-SEM microscope, CLEM*Site*-EM interfaces a commercial software (SmartSEM and ZEISS Atlas 5 from Carl Zeiss Microscopy GmbH) through a specific application programming interface (API) provided by Zeiss. The algorithms and high-level control functions that we developed for CLEM*Site* are openly accessible and free to download from a GitHub repository (see link in Materials and methods, Software availability).

### Correlation strategy

The correlation strategy applies transformations to translate cell positions (microscope stage coordinates) from LM into cell

positions of the FIB-SEM (Fig. 1 b). At the light microscope, cells of interest can be selected either by manually screening or using more assisted pipelines, such as the ones described in the application examples below. In our experiments, the Golgi apparatus morphology was used to select cells employing an automated phenotypic screen. At each position where a cell of interest is identified for downstream CLEM analysis, a light microscopy acquisition job is programmed to collect a set of images. The first set comprises one fluorescence image at low magnification (using a 10× objective, NA = 0.4; Fig. 2 a), and one reflected light image of the same field of view revealing the grid pattern (Fig. 2 b). The target area, which can be a cell or more precisely a subcellular region (e.g., the center of mass of the Golgi apparatus, Fig. 2 a), is placed at the image center. With a target centered, the stage coordinates are recorded for subsequent use in the correlation.

All images are then loaded to CLEM*Site*-LM. The first step of the software is to automatically extract landmarks that will be used as references to register the stage coordinates coming from LM and EM images. The grid pattern imprinted on the bottom of the culture dish is a convenient coordinate system for registration. As the screened cells are typically distributed across the whole surface of the CLEM dish, a map of local landmarks is built from multiple sparse images of the grid.

Since the bars constituting the grid are relatively thick at 40 μm wide, the center of their intersections is used as a fiducial marker. In CLEM*Site*-LM, these centers are identified by a line detection algorithm, which is applied to the reflected light images to find the lines present at the grid bar edges. At the grid bar crossings, the detected grid bar edges intersect in four points, the centroid of which is used to mark the center of each grid bar crossing (Fig. 2 b and Fig. S1). This center point is saved in stage coordinates as a landmark. Since each grid square is already imprinted with a unique combination of alphanumeric characters, each calculated center point is labeled using this existing identifier. Identification of the corresponding alphanumeric set of characters in reflected light images is performed by a VGG16-based convolutional neural network (CNN; Krizhevsky et al., 2017; Fig. 2 b). The CNN was trained with a combination of synthetic and manually annotated light microscopy images.

The last step in CLEM*Site*-LM is to obtain a second collection containing the centroid stage coordinates of the target structures (e.g., the Golgi apparatus, Fig. 2 a). In our experiments, since our target cells are centered on the image, stage coordinates are extracted directly from the image metadata.

After sample preparation for EM, removal of the coverslip, and coating with a thick layer of gold, samples are transferred to the FIB-SEM chamber, where they are left to equilibrate for 1 d before starting the experiment. The next day, the examined sample is positioned for optimal visualization of the grid (see Materials and methods, Correlation in EM). In the beginning, CLEM*Site*-EM requires an image from a random initial position of the sample surface to be used as a calibration step. The *Navigator* module prompts the user to indicate which grid square (identified by the alphanumeric identifier) is in the SEM image and in which orientation. The landmarks are then detected by the same line detector used by CLEM*Site*-LM. As a

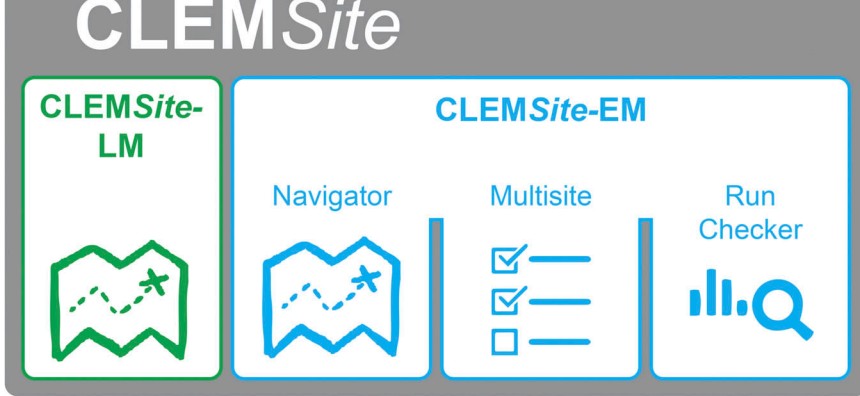

Figure 1. **Schematic representation of the correlative light and electron microscopy software CLEM*Site*. (a)** Overview of the different elements of CLEM*Site*, CLEM*Site*-LM, and CLEM*Site*-EM. CLEM*Site*-EM is divided into three modules: the *Navigator*, which allows to store and move to different positions in the SEM, then *Multisite*, which drives the FIB-SEM acquisitions, and the *Run Checker*, which controls and reports during the FIB-SEM runs. **(b)** Workflow for the automated acquisition of multiple correlated datasets. Light microscopy is performed to find specific phenotypes ("LM phenotyping"). From them, individual cells are selected ("LM targets") and their corresponding landmarks and positions are recorded using CLEM*Site*-LM. (i) This scheme illustrates that for "LM targets," the low magnification overview shows the selected cellular targets (green circles), the landmarks (pink circles) used for correlating across imaging modalities, and the alphanumeric coordinate system that is patterned on the cell culture dish. On the right, a higher magnification image shows more clearly the Golgi as the cellular target (green circle), and the landmark used (pink circle) provided by the patterned culture dish, whose position is referred to the closest alphanumeric coordinates of the culture dish. (ii) Inside the FIB-SEM, "EM targets" refers to the process of obtaining the positions of the cells in the EM (stage coordinates). For that, a transformation matrix T is calculated based on the respective landmark positions of LM and EM (LM landmarks list in pink and EM landmarks list in black). This matrix transforms an LM Target list (cell positions in LM stage coordinates in green) into an EM target list (cell positions in EM stage coordinates in orange). On the right, the blue trapezoid and rectangle represent the milled and targeted region on the surface of the sample, inside the

FIB-SEM. The black circle indicates the target coordinates in EM for the landmark, which should have its equivalent pink circle on LM stage coordinates. All of this correlation work is performed using the *Navigator*. (iii) Finally, in "FIB-SEM acquisitions," cell image volumes are acquired at the "EM target" positions using *Multisite* and *Run Checker*. At each location of interest, the focused ion beam (red arrowhead) and the electron beam (blue arrowhead) are iteratively used to acquire datasets. The acquired data is finally analyzed to characterize different phenotypes ("EM phenotyping").

fail-safe, landmarks can also be manually identified by clicking over them.

Based on the culture dish manufacturer's known grid layout (consisting of letters and numbers) and four landmarks, the software creates a linear model that represents a simple quadratic lattice to predict the position of all landmarks in stage coordinates. This preliminary model-based prediction of landmark positions has a targeting accuracy of ∼5 ± 20 μm (measured as root mean square error [RMSE]), which is insufficient for precise localization of the cell and therefore requires additional refinements. This involves obtaining more landmarks over the sample surface. Thus, at each predicted landmark, an SEM image is automatically taken, and a U-net-based CNN (Ronneberger et al., 2015) is used to compute the probability of each image pixel being part of a grid bar edge (Fig. 2 c). The line detector is applied again to the resulting grid bar edges to give the center point. This process is repeated throughout the sample surface to find and associate each landmark identified previously in light microscopy images.

When enough landmarks are collected, an affine 2D transformation is computed to register the landmarks from LM and EM. The transformation is applied to all LM stage coordinates of target cells to predict their position in SEM stage coordinates at the surface of the resin block (Fig. 2 d). When all four experiments are taken into consideration, this global transformation reduces the error in target accuracy down to 13 ± 6 μm. If the grid pattern is sharp and the block surface does not present any defects such as cracks, scratches, or dust, grid edges are detected perfectly, and the center point of the landmark can be calculated with higher accuracy (Fig. S2). In our case, we had two such experiments, reaching a global targeting accuracy (RMSE) of 8 ± 5 μm.

A local transformation delineates the third and final targeting refinement to further increase the targeting accuracy. It is calculated before each FIB-SEM acquisition, using only the landmarks close to the target (a total of eight landmarks falling in a radius of 1,200 μm). By applying this local refinement, we obtained a final targeting accuracy of 8 ± 4 μm for all the experiments (average of *n* = 10 cells per experiment over N = 4 experiments), or of 5 ± 3 μm with the pristine blocks (average of *n* = 10 cells per experiment over N = 2 experiments). These results were validated by registering manually the fluorescence image and the SEM view of the sample surface in the predicted position (Fig. 2 e and Table S1).

Thus, with our experiments, we exemplify how it is possible to perform automated detection and registration of landmarks from both LM and SEM imaging modalities, which can lead to a final correlation with an accuracy of targeting close to 5 μm. Besides, the correlation can be performed over relatively large sampling areas: in the experiments, a surface region of ∼8 × 8 mm² was completely mapped.

## Automation of FIB-SEM imaging of multiple cells

Once the correlations between cell positions in light and electron microscopy have been determined, the *Multisite* module of CLEM*Site*-EM executes the FIB steps of our automation workflow. The following steps, usually performed by a trained human operator, are triggered autonomously: localization of the coincidence point, needed to bring the FIB and SEM beams to point at the same position (Fig. 3 a); milling of the trench to expose the imaging surface and detection of the trench to ensure a well-positioned imaging field of view (FOV; Fig. 3 b); automated detection of image features in the imaged surface needed to find an optimal location for the initial autofocus and autostigmation (*AFAS*; Fig. 3 c); and finally the stack acquisition (Fig. 3 d). These four steps are executed sequentially for all targets (Fig. 3 e).

The sample is positioned at the target coordinates of the first cell, and the *Multisite* module performs the coincidence point alignment of both the electron and ion beams, a step which, in a typical acquisition would be carried out manually (Fig. 3 a and Fig. S3 a). To preserve the target from the burning radiation of the mark, the sample is shifted 50 μm in x. The working distance is checked by autofocus and adjusted by the z-movement of the stage. A square fiducial area (20 × 20 μm²) is then created at the surface of the block by FIB sputtering at a high current (7 nA). This square is then imaged by FIB and SEM sequentially (using the SE detector). The offset (in the y-direction) between the center of the sputtered square (i.e., the focus point of the ion beam) and the center of the e-beam image is then utilized to calculate the z-offset by applying a trigonometric relation (Fig. S3). A further refinement is achieved by cross-correlating images of the sputtered mark captured using the FIB (imaging current, 50 pA) and SEM modes. The measured difference in micrometers is then applied to the SEM beam shift to correct the FOV position.

Following the automated coincidence point alignment, the software proceeds with estimating the position of the target cell using the local transformation based on the closest landmarks as described above. After moving back to the estimated position, the software automatically triggers ZEISS Atlas 5 to mill a trench, which exposes a cross-section orthogonal to the surface of the block. When the milling is finished, an SEM image is taken with the ESB detector at a FOV of 305 × 305 μm², and the trapezoid shape of the trench is detected using thresholding and shape recognition (Fig. 3 b and Fig. S3 b).

The center of this shape is used as a reference to position the FOV to be imaged during volume acquisition. The FOV is then changed to 36.4 × 36.4 μm² to capture an image of the cross-section at higher magnification (Fig. 3 c). A feature detector (Harris Corner detector [Harris and Stephens, 1988]) is applied to this image to identify salient points with high contrast and complex pixel neighborhoods. Such point features usually cluster around complex cellular structures; therefore, they can

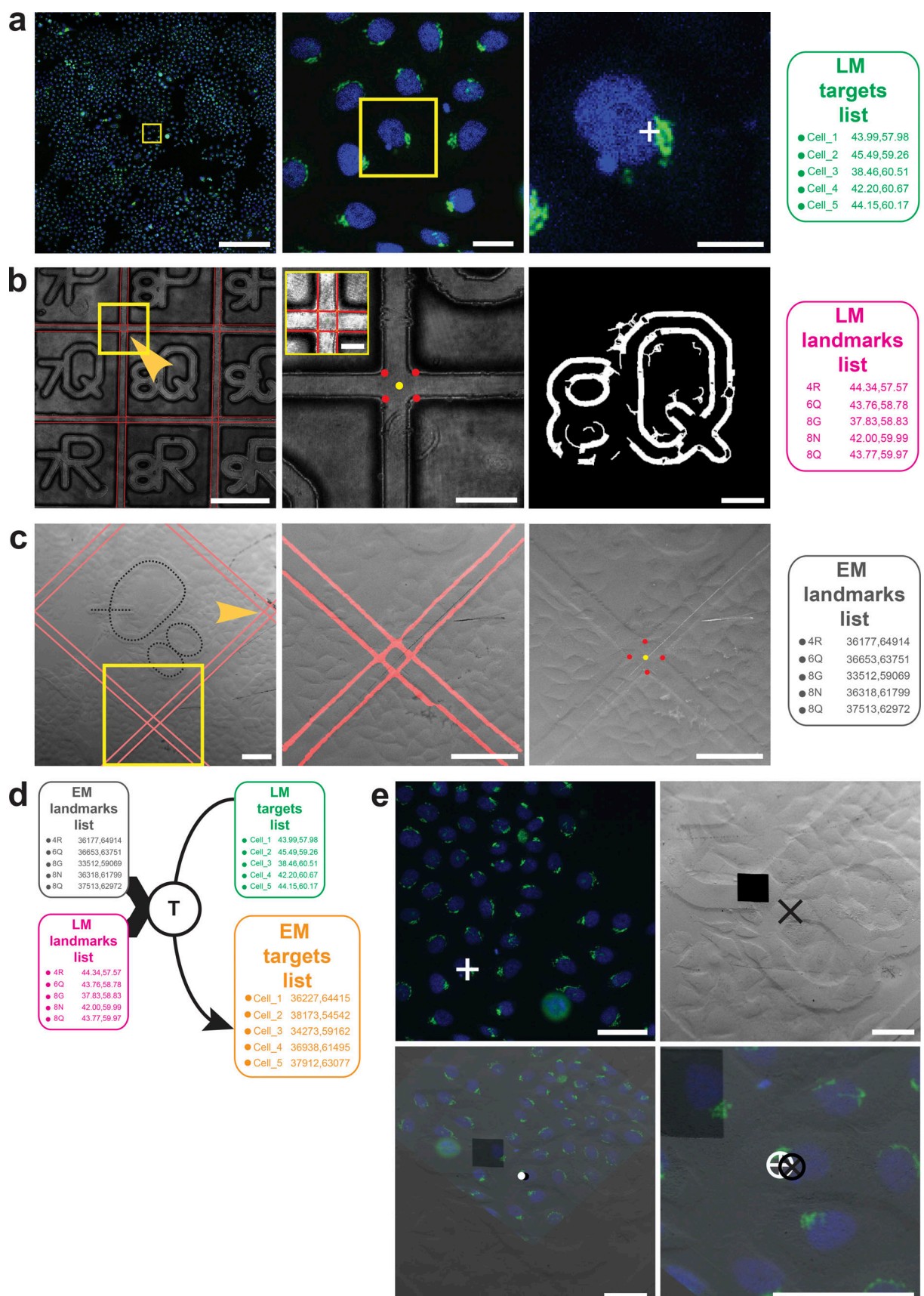

**Figure 2. Coordinate system mapping and automatic detection for the correlation strategy. (a)** Cell of interest selected using fluorescence microscopy by scanning low magnification images (first and second image). In our experiments, we targeted the Golgi apparatus center of mass (a, third image, white

cross). The image position is translated to stage position coordinates and stored in the "LM targets list" (green). **(b)** Simultaneously, reflected light images (b, first image) are stored, and later used to extract the stage coordinates of landmarks (*LM landmarks list, pink*). The image is analyzed and a line detector is applied (red lines). The intersection of the lines is used to find grid bar crossings (b, second image including inset). The corresponding detected edges are converted to lines that automatically mark 4 points (b, second image, red dots). Those points are used to determine the center point (second image, yellow dot), and they will be part of the "LM landmarks list." By convention, the top left corner (yellow arrowhead) is named by associating its unique center point (yellow dot) with the alphanumeric identifier imprinted onto the glass dish bottom. To identify the alphanumeric character on the image, the reflected light image is automatically thresholded and cleaned (b, third right image) using a combination of traditional image analysis pipelines (see Fig. S1) and then passed through a convolutional neural network for classification, in this case, 8Q. **(c)** In the FIB-SEM, the strategy of mapping is repeated: scan images are taken by the *Navigator* module (c, first image), and the grid bar crossings are detected to calculate the center point (red marks). In SEM, it is difficult to do automatic detection of the alphanumeric character (indicated by a dotted black line, not the process of automatic detection). For this reason, the first character must be identified by the user and then given as input to the map. Each grid bar crossing surrounding the character is imaged (yellow remark at the bottom). Here, a different convolutional neural network is used to evaluate the probabilities of being a line on each crossing (c, second image, red marks). The identification of the center position of the crossing is very similar to the one in LM, here the intersections (c, third image, red dots) are identified after line detection, and the center point is stored as a position (c, third image, yellow dot). This process continues at each predicted landmark to give a list of landmarks (*EM landmarks list*). **(d)** A transformation is computed to register the positions from the LM and the EM landmarks lists (pink, black), which is then applied to the *LM targets list (green)* to predict the respective *EM targets list (orange)* across the sample at the FIB-SEM. **(e)** At the end of the experiment, the position of the cell can be validated using manual registration. FM (first image, top left) and SEM (second image, top right) images were superimposed manually using the cell contours. For this, the FM images were flipped, rotated, and scaled (first image, bottom left). The position of the LM target (white cross) is then compared with the predicted target in the SEM (black cross) (second image, bottom right). This overlay of SEM with LM images was repeated for each experiment, obtaining a final targeting accuracy of 5 ± 3 μm (RMSD over *n* = 10). Scale bars: (a) 200, 25, 25 μm; (b) 200, 100 μm with small window upper left corner 25, 50 μm; (c) all 100 μm; (e) all 50 μm.

be clustered using k-means. The k-means centroids are additionally filtered and prioritized by higher variance, high entropy, and their proximity to the center of the image. The first element in the filtered list can thus be stored for the subsequent application of autofocus and autostigmation procedures (*AFAS*; Fig. 3 c). In the absence of a cell on the cross-section, the *AFAS* function is automatically targeted to the edge between the cross-section and the surface of the block. An image stack is then acquired (Fig. 3 d). The dimensions of the image stack, as well as the z resolution, are set when initializing the run, through the CLEM*Site* interface (Fig. S4 a). Whilst every cell of one run can be acquired with the same recipe (as defined in ZEISS Atlas 5 in sample preparation, where total volume to be acquired, slice thickness, and FIB currents applied are defined at each step), CLEM*Site*-EM also offers the individual definition of recipes, allowing a per cell adaptation of the shape or volume.

The *Run Checker* module of CLEM*Site*-EM (Fig. S4 b) supervises each stack acquisition and corrects the position of the FOV if an image drift occurs in the y-direction. Similar solutions were presented in (Marturi et al., 2013) and (Jin and Li, 2015), but in ours, the drift is computed using ASIFT point feature correspondences (Lowe, 2004; Yu and Morel, 2011), which are optimally filtered using RANSAC (Fischler and Bolles, 1981). When a drift is detected, the next image is corrected accordingly by adjusting the SEM beam shift. If the difference between slices is too big that is not possible to detect enough SIFT point features to align them, the image is aligned with respect to the gold coating on the top part of the image. *Run Checker* also continuously monitors the run for periodic autofocus and stigmatism. For each image acquired, Vollath's autocorrelation and a Laplacian metric (Pertuz et al., 2013) are used to measure, respectively, the quality of focus and stigmatism. When these values differ more than 25% between two consecutive slices, a warning message in the user interface and an e-mail is automatically sent to the user, who can then decide to interfere and correct the drift manually.

After completion of one volume acquisition, CLEM*Site*-EM restores the original microscope conditions, drives the stage to

the next target cell (using the *Navigator* module), and starts a new FIB-SEM run (*Multisite* module). This process is repeated until all targets are acquired (Fig. 3 e). When the Gallium source no longer produces a coherent ion beam, the FIB interrupts the current run. Upon reheating the Gallium source, the run is then manually resumed to proceed with the next cells. For a typical FIB-SEM acquisition recipe at our microscope (as outlined below in case study 1), 15 to 20 consecutive cells can be acquired before it becomes necessary to reheat. Thus, CLEM*Site* provides a unique solution for the automated targeted 3D acquisition of multiple cells previously identified by light microscopy.

## Applications

We illustrate CLEM*Site*'s capabilities with two applications. In the first, the Golgi apparatus morphology of HeLa cells is perturbed with siRNA knockdowns by adapting a previously described solid-phase reverse transfection protocol (Erfle et al., 2008), where several siRNA knockdowns can be performed in a single experiment. This approach represents an efficient screening tool to identify specific genes involved in Golgi apparatus morphology.

In the second application, we illustrate a follow-up of this screen, where morphological perturbations of the Golgi apparatus are further evaluated by focusing on one of the siRNA treatments, i.e., knocking down the COPB1 gene expression. This treatment was chosen based on its prominent phenotype. Variable transfection efficiency leads to a heterogeneous distribution of the phenotypes. We address this heterogeneity with our CLEM approach in which the target cells are selected according to their phenotype as visible by fluorescence microscopy. Using such a phenotype-enriched selection of cells enables us to collect sufficient data for a morphometric evaluation at the EM level.

### Case study 1: Integrated multiple knockdown CLEM screen
Organelle morphologies can be observed by fluorescence light microscopy and used as a proxy to identify which genes are involved in various cellular functions. Previous experiments showed how the Golgi apparatus organization can be studied by

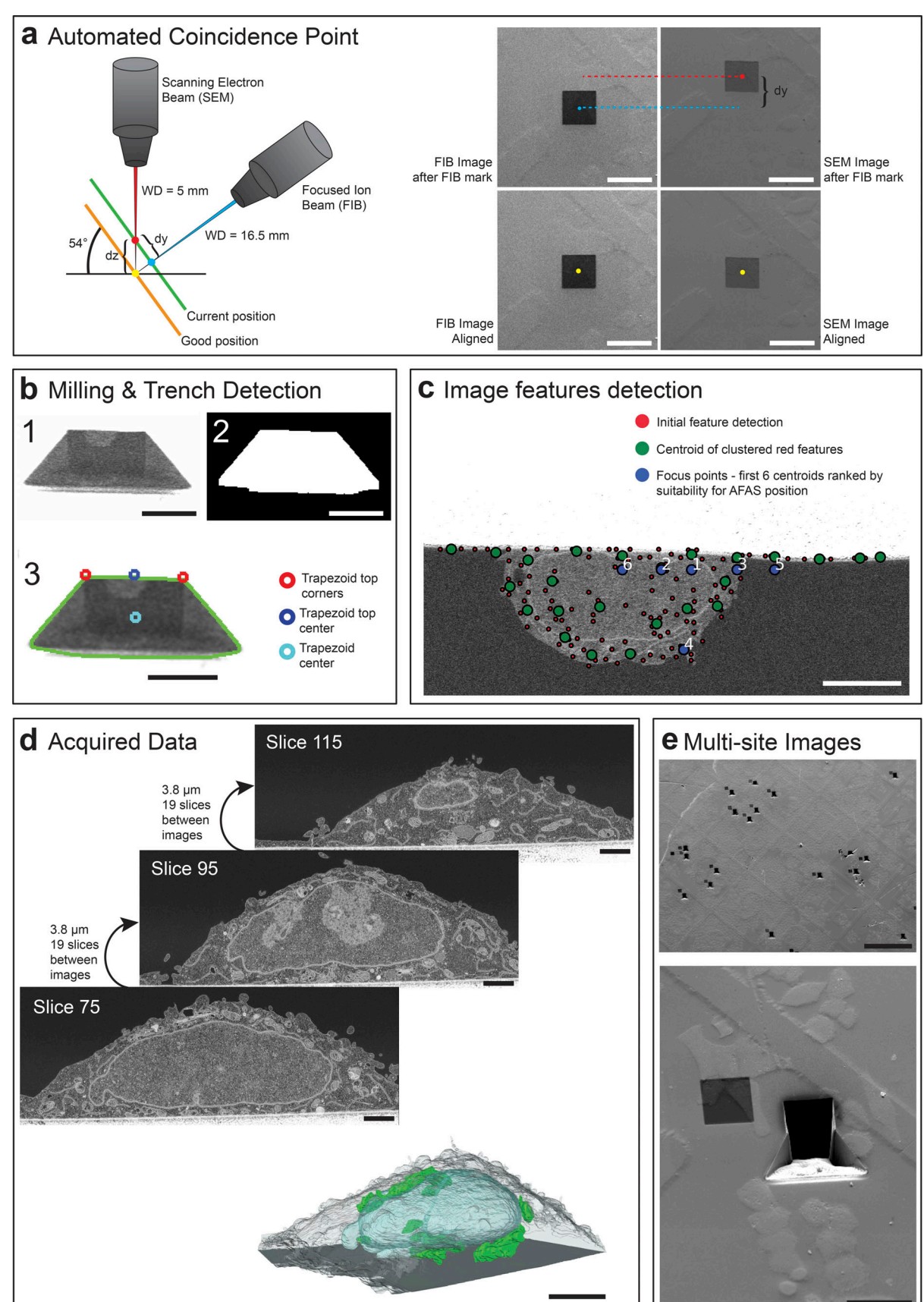

Figure 3. **Schematics of some of the implemented components to achieve FIB-SEM automation and its results. (a)** Automated Coincidence Point routine is illustrated schematically. When not tuned, the two beams are usually pointing at different positions of the sample surface (green plane, blue point for

FIB center, red point for SEM center). The orange plane below shows the case where the ideal position (yellow point) is achieved for both FIB and SEM beams. In the software routine, a square is sputtered with the ion beam on the sample surface. The offset between the two beams is calculated based on the difference between the center of the sputtered mark in the SEM and FIB images (dy, distance between red and blue positions in the green plane). The z height (dz) of the stage is then corrected, and a further refinement using the SEM beam shift is performed by calculating the translation of the square mark between FIB (50 pA image) and SEM images. **(b)** Milling & Trench Detection: (1) After finding the coincidence point, a trench is milled to expose a cross-section at the region of interest. (2) The trench is detected to accurately position the field of view. First, three-level thresholding is applied to the image, followed by the detection of the biggest connected component that fits a trapezoid shape. From the final binary shape, boundaries of the trapezoid are found (3): the top corners (red circles), the trapezoid top center (blue circle), and the trapezoid center (light blue circle). **(c)** Image features detection: The image of the cross-section surface is analyzed and scored for the best focus positions to perform autofocus and autostigmatism. Features inside the image are found by using Harris corner detection and the variance of a small region surrounding each detected corner position. The initial features (red points) highlight the high contrast and complex areas of the imaging surface which usually cluster on cellular structures. Features are clustered and their centroids (green dots) are then filtered and prioritized to detect the first 6 ones suitable for *AFAS* (blue points). Due to the brightness/contrast settings to make the cell visible well inside the cross-section, the top surface of the sample above the cellular edge, which is covered with a gold coat, is only faintly visible. This region is excluded from the analysis of the cross-section to prevent autofocus outside the proper field of view. **(d)** Acquired data: Images are acquired at 200 nm intervals (in z) throughout the Golgi apparatus region. The resulting stack is used for 3D render and quantifications. **(e)** Multi-site images: Result of an experiment, where multiple targets had been acquired automatically across the full surface of the sample. Scale bars: (a) all 50 µm; (b) all 25 µm; (c) 5 µm; (d) slices all 2 µm, model 5 µm; (e) 500, 50 µm.

tagging GalNAcT2, a resident enzyme of the Golgi apparatus, with a fluorescent protein (Storrie et al., 1998; Simpson et al., 2007). To efficiently screen the effects of different knockdowns, we have adapted an integrated experimental approach based on solid-phase reverse transfection (Erfle et al., 2008). By depositing drops of siRNA transfection mix, multiple treatments are distributed as an array at the surface of one single gridded culture dish. With such a layout, up to 32 spots can be deposited (Fig. 4 a).

After a 72-h incubation period, the cells on each siRNA spot are automatically imaged by confocal fluorescence microscopy. For this, four fields of view in each treatment spot are imaged with a 10× objective (NA = 0.4, Leica HC APO). The position of these fields is generated systematically using a matrix pattern. The resulting fluorescence images are then processed in Cell-Profiler (Carpenter et al., 2006), where the nuclei (DAPI channel) and a total of four features associated with the Golgi apparatus (GFP channel) from individual cells are extracted. Upon perturbation of the secretory pathway, the Golgi apparatus morphology can display a variety of phenotypes (Simpson et al., 2012), which we classified into four typical appearance categories: fragmented, diffuse, tubular, and condensed (Fig. S5 a). We designed the four features to score each one of such morphologies individually (fragmentation, diffuseness, tubularity, and condensation) to measure the impact of each siRNA treatment (Fig. S5 b). Thus, a high score on one of the features serves as an indicator of the presence of the phenotype.

For this proof-of-concept experiment, the expression of 14 genes was challenged (Table S2). The most striking effects were observed when perturbing the expression of subunits of the COP1 complex, associated with non-clathrin-coated vesicles (Fig. 4, b and c). For the three subunits tested (COPB1, COPB2, and COPG1), a considerable number of cells started to display a diffuse GalNAcT2-GFP signal, as visible by fluorescence microscopy after 72 h of treatment (Fig. 4, b and c). Under these experimental conditions, the other gene knockdowns did not display noticeable phenotypes (Fig. S5 b), likely because the sample size was not big enough to detect subtle variations in the Golgi morphology.

Applying our automated CLEM workflow, we selected two to three cells per condition for further ultrastructural analysis by

FIB-SEM. A total of 34 cells were automatically targeted (plus two control cells acquired manually) and acquired across three runs. For treatments with siRNA perturbing the expression of subunits of the COP1 complex, the cells were chosen from the pool that displayed the highest diffuseness score (Fig. 4 b, cells highlighted as triangles on the plot), a pool that was clearly distinguishable from the control condition. For other genes, even though the image analysis did not reveal any outstanding subpopulation, we picked randomly between the cells displaying the highest scores associated with the expected phenotype, as hypothesized from previous experiments (Fig. S5 b, selected cells highlighted as triangles).

At the EM level, five out of the six cells treated with COPB siRNAs with a diffuse phenotype displayed total disruption of the Golgi stack, which would normally display three to four closely associated cisternae. Instead, the region with enriched GalNAcT2-GFP fluorescence signal was filled with numerous vesicles (50–300 nm in diameter), suggesting a complete disassembly of the Golgi stacks upon knocking down the COPB1, COPB2, or COPG1 genes (as observed in the COPB1 of Fig. 4 c). For the remaining cell, a mixture of Golgi stacks and vesicles was observed.

The selected cells from the other siRNA treatments (Table S2) were also imaged by FIB-SEM to detect any subtle perturbations of the Golgi morphology at the ultrastructural level. For each condition tested though, the Golgi apparatus was visible and a stereological analysis (Ferguson et al., 2017) of the stack composition or stack volume did not reveal any differences with respect to the control (Fig. S5 c).

Altogether this experiment shows that our software can be utilized to screen for cellular and subcellular phenotypes in a large-scale CLEM experiment. When used in an integrated experiment with different siRNA treatments, CLEM*Site* enables automated and fast screening for protein knockdown effects on the fine ultrastructure of the Golgi apparatus.

### Case study 2: Screening for phenotypes
Specific gene knockdowns lead to perturbed phenotypes of the Golgi apparatus. As shown in the previous experiment, a striking phenotypic change occurs when cells are treated with siRNAs targeting subunits of the COP1 complex. Integrated screens

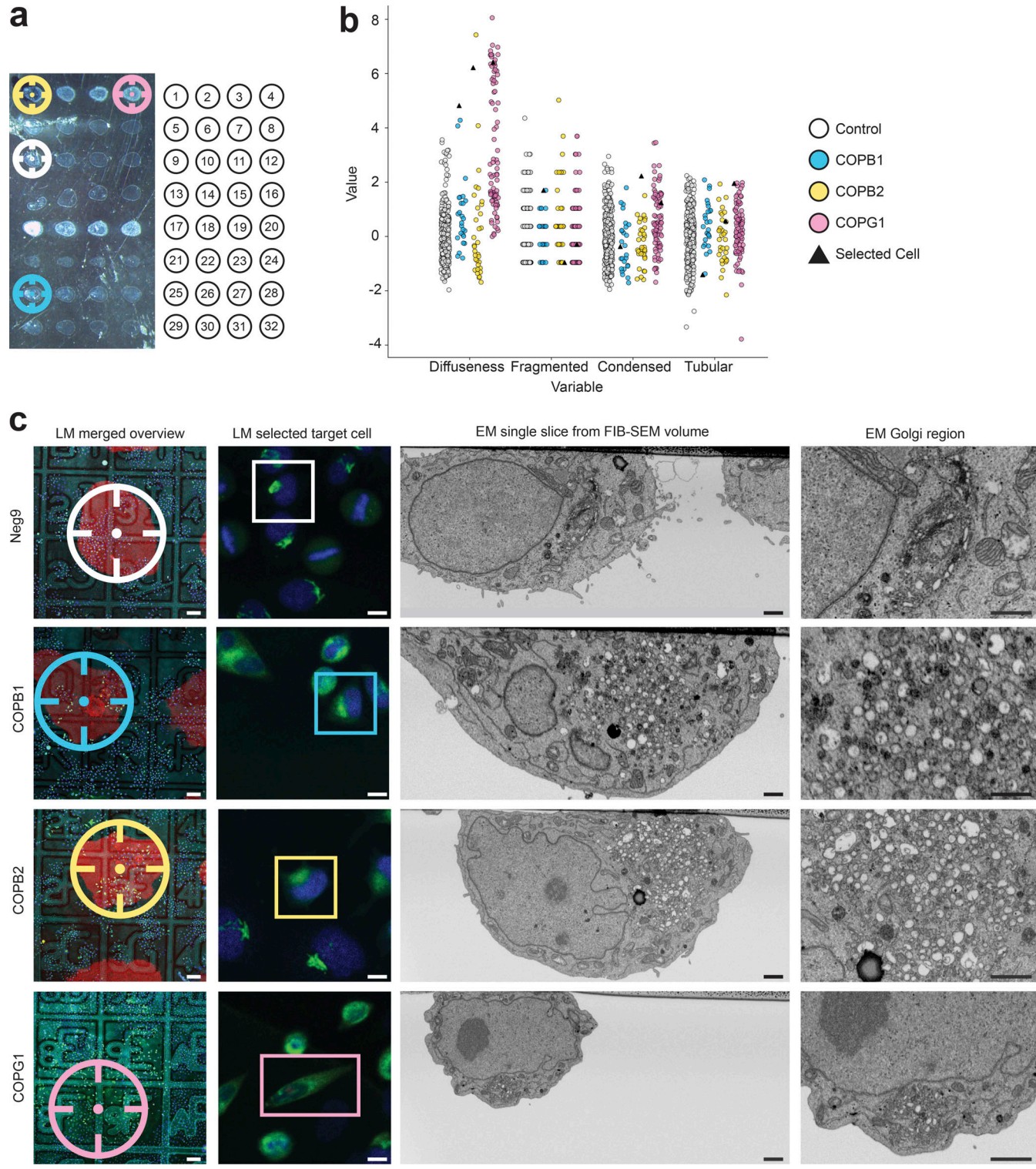

Figure 4. **Automated screen of 14 siRNAs after 72 h solid-phase transfection knockdown. (a)** Transmitted light image of one Petri dish with the 32 siRNA spots (left), where each siRNAs transfection mix is placed in the culture dish following a definite arrangement, see Table S2 for further details (right). **(b)** Morphological features of the Golgi apparatus scoring tubularity, diffuseness, fragmentation, and condensation for COPB1 (n = 26), COPB2 (n = 34), COPG1(n = 88) in comparison to negative control (Neg9, n = 305). Values of each feature are normalized with respect to the mean of the control. During the light microscopy workflow, cells transfected with COP siRNAs display a phenotype that can be identified because of their high value in diffuseness. As an example, we selected one cell of each COP-related siRNA (black triangles), to display in (c) the final result of the correlative experiment. **(c)** Selected correlated cells control (Neg9), COPB1, COPB2, and COPG1 (top to bottom): overview merged fluorescent, reflected light image and image of the siRNA spot (LM merged overview), the fluorescent image of a selected cell (LM selection target cell), a cross-section through the selected cell in the region of the Golgi apparatus acquired automatically with the FIB-SEM (EM single slice from FIB-SEM volume) and a zoom into the Golgi region (EM Golgi region). Three corner siRNA spots are highlighted with fluorescent gelatine (Alexa 594), shown as a red outline, whereas the last corner siRNA spot is highlighted with gelatine (Oregon green) shown as a green outline to make the orientation always recognizable. Scale bars: (c) left to right, 100, 10, 1, 1 μm.

with several treatments provide a reduced surface area where cells are exposed to siRNA. This in turn limits the number of phenotypic cells accessible for each condition. Therefore, we performed a second experiment, where the entire cell population of a culture dish was exposed to the treatment. We focused on a COPB1 siRNA treatment by liquid-phase transfection and evaluated it after 48 h of incubation. Although a larger number of cells displayed a diffuse phenotype under these conditions, the observed phenotypic diversity justified the use of CLEM to perform an ultrastructural analysis on the most perturbed cells.

As described above, a measure of cytoplasm fluorescence intensity levels was used as a score to select the diffuse phenotype. By defining a threshold on this score, all cells with a high value of cytoplasmic diffusion were selected and then the diffusion phenotype was validated manually for each cell using a customized Jupyter notebook (see Materials and methods). Using adaptive feedback microscopy (Tischer et al., 2014), the identified target cells were automatically re-imaged on the LM, acquiring the image sets necessary for the correlation (reflected light and confocal fluorescence at 10× magnification, see Results, Correlation strategy). Higher magnification z-stacks of the cell and Golgi apparatus were also acquired with the 40× objective (zoom factor ×4) to document the spatial distribution of the organelles. The 3D information acquired here was valuable, for example, to be registered to the 3D FIB-SEM volumes (Fermie et al., 2018).

In the next step, the set of LM images was processed as described previously, to establish a list of LM landmarks and a precise list of target cell locations. The cells were prepared for EM and transferred to the FIB-SEM where CLEM*Site* autonomously acquired image stacks at each target location. In the example shown in Fig. 5, the LM screen resulted in the selection of 90 cells. Given the prototypic nature of our workflow, we kept this initial number higher to compensate for the loss of targets when progressing downstream. A first selection removed the cells that were too close to each other (<150 μm) or that were on regions damaged during sample preparation (resin defects, scratches at the surface of the block; see Materials and methods, Correlation in electron microscopy). Following this filtering step, a final selection of 30 to 40 cells was acquired as FIB-SEM stacks. After this, we examined the automatic acquisitions to get the ones that had sufficient quality for analyzing the fine morphology of the Golgi apparatus. Common criteria to discard acquired stacks were out of focus during acquisition due to technical failure, partially acquired samples due to targeting inaccuracies, or non-valid samples due to damages on the sample surface, cells being multinucleated or mitotic (see Tables S3 and S4).

Altogether, on average, around 20 cell volumes per experiment were acquired and analyzed (Fig. 5 a and Videos 1 and 2) over an automated run that lasted 8 d, including one required stop for manual reheating of the gallium source. Note that these cells were distributed across a 40 mm² surface area with a maximum distance of 8.2 mm between cells. Our program fully automatically and efficiently performs correlations between fluorescence microscopy and FIB-SEM data. As an example, the rendered segmentation of the FIB-SEM volume perfectly recapitulates the cell morphology seen in FM (Fig. 5 b), demonstrating the accuracy of the correlation. The resolution of the FIB-SEM images is sufficient to analyze the ultrastructural details of numerous cells. In our case (COPB1 knockdown) we could reveal how the Golgi complex transitions from a stacked organization to an accumulation of vesicles (Fig. 5 c).

## Discussion

Performing CLEM on cultured cells, after selecting cells based on their phenotype during the LM step, is an efficient way to achieve ultrastructural analysis on targeted subpopulations. This selection is performed before the EM sample preparation and often one cell at a time. Because of an unprecedented efficacy to interrogate the cell ultrastructure in 3D, volume SEM imaging (Titze and Genoud, 2016) is gaining popularity in the life sciences. Volume SEM has been used in CLEM experiments to capture phenotypic cells in culture (Mellouk et al., 2014; Russell et al., 2017; Ohta et al., 2021). In most cases, volume SEM and CLEM were combined to capture the 3D ultrastructure at the highest spatial resolution possible (isotropic for FIB-SEM). Yet, the resulting low imaging throughput, in combination with individual cell picking, previously rendered volume SEM impractical for ultrastructural screens. Only in rare cases, several cells were analyzed in a single experiment (Cosenza et al., 2017). Consequently, volume CLEM is rarely used for screening large populations of cells.

Using the novel workflow we developed, it was shown that correlative imaging using FIB-SEM can acquire multiple targets within a single experiment (up to 30 over 1 wk of acquisition) with full automation. Detection of local landmarks imprinted in the culture substrate enables automated correlation and targeting with a 5 μm accuracy. We estimate that this number could still be improved by customizing a gridded substrate with a smaller mesh size, consequently, shortening the distance between landmarks and targets. Our detection algorithm could be extrapolated to other customized dishes or commercial substrates for cell culture in SEM samples (Luckner and Wanner, 2018b). An advantage of using local landmarks for the correlation is that they mitigate the impact of sample surface defects or optical aberration across long distances. Alternatively, targeting individual cells with a FIB-SEM has been achieved by mapping the resin-embedded cells with microscopic x-ray computed tomography (Hoffman et al., 2020). We speculate that such tools could be an alternative to a gridded substrate yet cannot predict its adaptability to large resin blocks such as the ones we used in this study.

Thanks to the utilization of a FIB-SEM, nearly the whole sample surface is accessible, enabling the correlation of multiple cells. In the case of highly distributed and distant rare events (Guérin et al., 2019), the respective targets are still within reach. We demonstrate the workflow on commercial dishes with a usable surface on the order of 40 mm², but much larger surface areas are possible. The limitation is dictated mainly by the dimensions and travel range of the microscope stage. With such potential, the other main feature of our software is the ability to trigger an autonomous acquisition in multiple sites in one

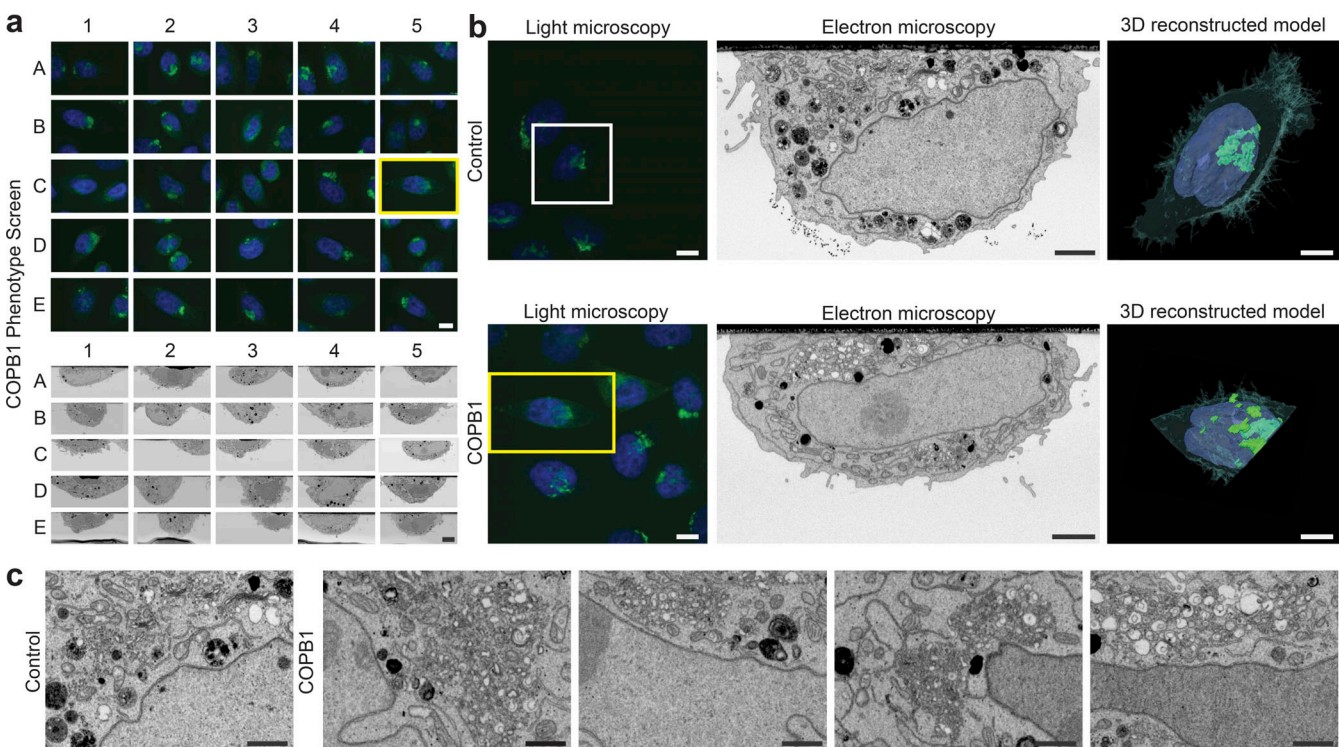

**Figure 5. Automated screen on COPB1 cells in light and electron microscopy 48 h after liquid phase transfection knockdown. (a)** Overview of 25 selected cells in a screen for COPB1 knockdown. Light microscopy images (green is GFP GalNAc-T2 Golgi apparatus and blue is DAPI for the nucleus, top) and the corresponding electron microscopy images (bottom). **(b)** Top: Selected control cell (treated with XWNeg9 siRNA) in light microscopy (left), electron microscopy (middle), and a reconstructed model from the FIB-SEM stack (right) showing the 3D model of the nucleus in blue, the model of the Golgi stacks in green and a surface rendering of the cell surface in transparent green. Bottom: Selected COPB1 cell (treated with COPB1 siRNA) in light microscopy (left), electron microscopy (middle), and a reconstructed model (right). **(c)** Detailed electron microscopy images of the Golgi apparatus region in a control cell (left) and four different variations of a disturbed Golgi apparatus in different selected cells of the COPB1 knockdown. Scale bars: (a) LM—10 µm, EM—5 µm, (b) left to right—10, 2, 5 µm, (c) 1 µm.

microscopy session. This fully automated triggering has not previously been achieved on biological samples. This is made possible through the automation of key steps of the imaging pipeline, i.e., (1) setting the coincidence point of both ion and electron beams, (2) automated selection of key focusing ROIs using computer vision, and (3) constant tracking of the sample position within the field of view of the microscope. In summary, all interactions with the microscope that are usually supervised by a human operator during acquisition, be it several hours or days, are automated.

One essential strategy for increasing the acquisition throughput is the decision to decrease the resolution in the z-dimension, thus prioritizing the speed of acquisition and ultimately the total number of cells acquired in one run. For many of the morphological features used, low z resolution has been proven efficient to score phenotypic variability at the subcellular level (Lucocq and Hacker, 2013). Here, the images in one volume are acquired every 200 nm, a step size much larger than typically used for isotropic voxel acquisition (e.g., 4–8-nm resolutions). The resulting gain in speed is significant, leading to only 6 h necessary to acquire one full cell (including the creation of the trench). This is in stark contrast to isotropic acquisitions that can take from days to weeks per adherent cultured cells (Xu et al., 2017; Luckner and Wanner, 2018a; Hoffman et al., 2020).

Extrapolating acquisition time to a screen of about 30 cells, our workflow can deliver results in 10 d compared to isovoxel acquisition regimes that would require more than 6 mo of machine time. Therefore, CLEM*Site* is intended to be the foundation of a screening tool for performing quantitative assessments of morphological variations. It could help to reveal rare or novel phenotypes at the ultrastructural level, and simultaneously increase the number of observations. Other acquisition regimes of FIB-SEM can be considered if higher resolutions are required, but at the cost of a (much) lower throughput.

We believe that other research questions could benefit from this type of screening. As an example, the Human Protein Atlas Image Classification competition (Ouyang et al., 2019; Le et al., 2022) managed to classify multiple organelles of individual cells in fluorescence microscopy. Such machine-learning models could be used to find rare events or particularly interesting phenotypes. In another example, in host–pathogen interactions, early infected cells might start to display a recognizable phenotype in a small subpopulation of cells (Santarella-Mellwig et al., 2018). In both cases, those marked cells could be used to establish a FIB-SEM screening to discover new morphological differences at the micrometer level.

To expand the applicability of these screenings beyond the proof-of-concept here presented, we propose two directions of

improvement. First, by acquiring smaller enclosed volumes with isotropic resolution, we could target area-delimited organelles, like centrioles (Cosenza et al., 2017). In this case, the full cell volume is neglected in favor of a small portion of it, but with higher z resolution. At the software level, that would require improving the targeting accuracy using smaller grids and extending the maps to 3D coordinates. 3D registration against a light microscopy Z-stack would considerably help to constrain the field of view during acquisition (Loginov et al., 2022), thus reducing the imaging time and keeping the field-of-view position during tracking. At the instrument level, this would require, first, stabilizing the ion beam before the critical region is acquired, to compensate for the change between high currents for milling and fine currents for sectioning; second, to make sure that the fine current beam hits exactly the front face of the milled cross-section and then prevent milling artifacts; finally, the second direction is to increase the number of samples acquired per session. That would imply ion beams that automatically reheat the Gallium source when it is exhausted (like proposed in Xu et al. [2017]), with faster algorithms for autofocus and autostigmatism in SEM.

Capitalizing on the software's ability to screen across the full surface of the dish, we demonstrate that multiple siRNA treatments can be performed in a single-integrated CLEM experiment (by spotting siRNA onto the culture substrate). Provided that other treatment reagents can be bound to the culture substrate we anticipate that the same approach can be expanded to screening the effects of various drugs on subcellular morphologies. While we focus on enhancing existing hardware with targeting and automation abilities in this work, the next challenge is to efficiently analyze the resulting large amount of data generated. So far, we are using the powerful tools brought by stereology. We think that following the same principles, especially when designing the sampling strategies (Ferguson et al., 2017; Lucocq, 2008), the manual assessment of subcellular morphologies will be replaced by applying state-of-the-art computer vision, such as deep-learning-based semantic segmentation (Xu et al., 2021; Heinrich et al., 2021), followed by morphometric analysis. Once these tools are readily available, CLEM*Site* will be endowed with even more power to support molecular cell biologists in morpho-functional studies.

## Materials and methods

### Cell culture

HeLa cells stably expressing GalNAc-T2-GFP (Storrie et al., 1998) were maintained in Dulbecco's Modified Eagle's Medium (DMEM; Sigma-Aldrich) culture medium containing 10% fetal calf serum (Gibco Life Technologies), 100 units/ml penicillin (Gibco Life Technologies), and 100 µg/ml streptomycin (Gibco Life Technologies) and 2 mM L-Glutamine (Sigma-Aldrich), incubated at 37°C and 5% $CO_2$. Cell selection was applied using 500 µg/ml Geneticin (G-418 sulfate; Gibco Life Technologies) for every passage of the cells. Cells were incubated on gridded MatTek dishes (P35G-2-14-C-GRID, MatTek corporation) with siRNA spots and incubated for 72 h in DMEM medium without phenol red.

### siRNAs

siRNAs targeting Golgi apparatus (Simpson et al., 2012) morphology in this study were obtained from Ambion/Thermo-Fisher as Silencer Select reagents. Please see Table S2 for siRNA IDs and sequences.

### siRNA pre-screen and solid-phase reverse transfection

From an initial genome-wide screen for proteins affecting the secretory pathway (Simpson et al., 2012), 143 siRNAs affected the morphology of the Golgi apparatus. From them, 79 of the strongest phenotypes were selected. These were used in a pre-screen to find the most promising candidates for further CLEM experiments. 96-well plates (glass-bottom) were coated with siRNA transfection mixtures (Erfle et al., 2008). In brief, a 0.2% gelatine solution in $H_2O$, as well as a sucrose/Opti-MEM solution (1.37 g in 10 ml) is prepared and one transfection mix per siRNA composed of the sucrose/OptiMEM solution (3 µl), Lipofectamin 2000 (1.75 µl) and $H_2O$ (1.75 µl) is made. Silencer select siRNA (5 µl) of a 3 µM stock solution is mixed with the transfection mix, incubated for 20 min at room temperature and the gelatine solution is added (7.25 µl). The mixture is spotted onto the MatTek dishes and dried in a vacuum centrifuge at 37°C. Afterwards, HeLa Kyoto cells stably expressing GalNAc-T2-GFP (3,400 cells/well) were seeded using an automated cell seeding device (Multidrop/Thermo Fisher Scientific). Cells were imaged on a ScanR microscope (Olympus, UPlanSApo 20 × 0.7 Ph2, DAPI, GFP, and transmitted light). The plates contained control siRNAs for which the phenotype is well characterized on a light microscopy level: siRNA targeting COPB1, AURKB, KIF11, and nonsilencing negative control siRNA (XWNEG9). The 14 siRNAs showing the most prominent phenotypes were chosen for further CLEM experiments. Candidate selection was based on the morphology of the Golgi apparatus, as visible from the fluorescent signal given by the GalNAc-T2-GFP. From collected images, morphological features were computed as explained below (Light Microscopy prescan and CellProfiler feature extraction). Selected siRNAs were spotted onto a gridded MatTek dish (P35G-2-14-C-GRID) using a contact spotter (ChipWriter Pro-Bio-Rad Laboratories) resulting in a layout of 4 × 8 spots. The mixture either contained Oregon-green 488 gelatine (Thermo Fisher Scientific) or Alexa-494 gelatine (labeled with molecular probes protein labeling kit, Thermo Fisher Scientific) to make the spot boundaries visible. The array contained a total of six controls, as follows: three spots of negative control siRNA (XWNEG9), two spots of siRNA against AURKB and KIF11 (transfection control), and one spot of siRNA against COPB1. The other spots contained siRNA that target genes showing a Golgi phenotype after RNAi knockdown. 70,000 cells per ml were seeded onto the spotted MatTek dishes and fixed with a light fixation (0.5% glutaraldehyde, 4% formaldehyde in 0.1 M PHEM) after 72 h of siRNA treatment. The observed transfection efficiency for a successful experiment was not uniform and oscillated between 40 and 70% within the spots.

### Liquid-phase transfection for COPB1 knockdown

Liquid transfection with the siRNA (S3371) that is associated with the gene COPB1 was used in gridded MatTek dishes (P35G-

2-14-C-GRID) containing numbers and letters where cells were seeded at 70,000 cells/ml per dish. A standard protocol for transfection was used combining 3.3 µl of 30 µM siRNA with 1.5 µl of Lipofectamine 2000 (Invitrogen). Cells were examined by light microscopy 48 h after transfection.

### Fixation before light microscopy

Cells were fixed with a mixture of 4% formaldehyde and 0.5% glutaraldehyde (EM grade EMS) in 0.1 M PHEM Buffer (pH 6.9: 240 mM PIPES [Sigma-Aldrich], 100 mM Hepes [Biomol], 8 mM MgCl$_2$ [Merck], and 40 mM EGTA [Sigma-Aldrich]). A Ted Pella BioWave microwave with a temperature control unit (Pelco Biowave microwave with ColdSpot [Ted Pella, Inc.]) was used to accelerate the fixation process to 14 min at 250 W. DAPI (1 µg/ml in 0.1 M PHEM; Thermo Fisher Scientific) was applied to cells to stain the nucleus for a total of 10 min. To quench glutaraldehyde auto-fluorescence, cells were rinsed with 150 mM glycine (Merck) in PHEM buffer. Cells were left in the PHEM buffer for imaging.

### Light microscopy

#### Light microscopy prescan and feature extraction

After the MatTek dish was mounted on the light microscopy (LM) stage (Leica SP5 MSA), the four corner spots of the siRNA array (Fig. 4 a) were located based on their green/red fluorescence using a 10× objective (NA = 0.4, Leica HC PL APO 10×). At each corner, we used a python script to save the stage position from the microscope. After storing the positions of the four corners, the script generated a list of stage positions (2 × 2 sub-positions within each siRNA spot) that were loaded as positions onto the Leica Matrix Screener software. For the prescan images, the following specifications were used: 10× objective, 680 × 680 pixels, zoom 6, FOV 258 × 258 µm, 4x averaging, sequential scan for excitations 405 nm (DAPI-labeled nuclei, emission 457 nm), 488 nm (GalNAc-T2-GFP, emission 510 nm), and 594 nm (A594-labeled gelatine, emission 610 nm). Prior to each acquisition, autofocus was performed on the DAPI signal. A CellProfiler image analysis pipeline (http://cellprofiler.org/releases/, version 2.2) was configured to segment nuclei based on the DAPI signal and then delimit cytoplasmic cell ROIs by radial dilation of each nuclear ROI. Within each cell ROI, the GalNAc-T2-GFP signal was used to compute four intensity-independent features characterizing different typical alterations of Golgi morphology:

**Diffuseness.** Diffuseness was designed to characterize the fraction of the GFP signal dispersed in the cytoplasm. Given the image with the GFP signal, with the cytoplasm already segmented for each cell, the diffuseness of a cell is computed as the sum of all pixel values of the cell cytoplasm after a morphological grayscale opening, divided by the sum of all the pixel values of the cell cytoplasm. This value is high when the signal intensity of the Golgi is homogeneously distributed over the cell body.

**Fragmentation.** This feature was designed to characterize the number of unconnected Golgi structures. In some phenotypes, the Golgi apparatus is split into many pieces of variable size, with the biggest pieces being much smaller than a typical Golgi shape that would be observed in the negative control. Fragmentation is calculated by counting the number of separate connected components after a top hat morphological filter and Otsu thresholding (Otsu, 1979) of the GFP signal for each cell.

**Tubularity.** Some phenotypes showed high tubularity, also described as "enlarged" (Simpson et al., 2012), where the Golgi apparatus had elongated cisternae running through the cytoplasm. Morphological grayscale openings of the GFP signal are computed using structural elements in the form of a line at different angle orientations (0–180°). The difference between orientations that yields the maximum and minimum results of the filter are saved for each pixel. After this, the total value of elongation is computed as the sum of all values divided by the sum of the GFP intensity gray value for each cell.

**Condensation.** In other siRNA treatments, the Golgi was condensed in a smaller area, looking almost circular at the fluorescent microscope. Condensation was measured by the shape factor, which is calculated using the formula (4 * PI * Area) / (Perimeter$^2$) on connected components of a binarized image. The binarized image was obtained by performing a top-hat filter and an Otsu thresholding of the GFP intensity for each cell cytoplasm. If the length of the outline from the binarized object delineates a perfect circle, it will match with the area of the enclosing circumference, providing the maximum value of 1.

#### Cell selection using a Jupyter notebook

Images from the previous step were stored in separate folders (one for each position where images were taken), and the CellProfiler pipeline was applied to them to compute the features described in the previous step. The output values for the features were stored in a comma-separated value file and read with python into a Jupyter notebook (Kluyver et al., 2016; https://github.com/josemiserra/CLEMSite_notebooks). The first step, known as quality control (QC), was to remove possible artifacts or undesired effects like dividing cells or cells too dim to be properly classified. This step was also useful to explore the results by analyzing the values of the controls with respect to treatments and carrying on an exploratory analysis of our features. Pandas (McKinney, 2010) was used to read the files from the CellProfiler pipeline and organize the information associated with each cell in tables. Bokeh (https://bokeh.org/) and Holoviews (https://holoviews.org/) python packages enabled us to provide interactive plots inside the Jupyter Notebooks that increased usability and speed up the process of manual cell selection.

Once the feature values calculated in CellProfiler were loaded, cells expressing too little GalNAc-T2-GFP were rejected based on their integrated signal. Next, mitotic cells were rejected based on the coefficient of variation (CoV) of the DAPI signal, using the observation that, due to chromosome condensation, mitotic cells had a higher CoV than interphase cells within the segmented DAPI region. Finally, the *PowerLogLogSlope* feature (Bray and Carpenter, 2018) was used to remove potentially out-of-focus cells. After this, images of positive controls (AURKB, KIF11, and COPB1) were shown for qualitative assessment, and the experiment was continued if the three positive controls showed visible effects and the negative control was under standard conditions.

After the QC, which filtered out around 20–30% of the cells, the selection of cells for CLEM was displayed in a series of

interactive plots. The plots include controls that allow cells to be selected easily in groups. For each gene, the phenotype was differentiated from controls by one of the main features (e.g., COPB1 by diffuseness, ACTR3 by condensation [Fig. S5 a]), and cells were selected inside a jitter plot based on their feature values. Some phenotypes could manifest synergy in the feature space, for example, those that showed both fragmentation and tubularity. For this reason, t-SNE (Van Der Maaten and Hinton, 2008) was also proposed to cluster and select small clusters of populations naturally.

After the coarse selection of groups, small cropped preview images of each cell in the subpopulations for each gene are displayed. The user is prompted to individually confirm that the automatically selected cells exhibit the expected phenotype. The interface supports this through a yes/no button below each cell picture. A minimum of 42 cells were picked (three per treatment in a total of 14 treatments, the transfection controls AURKB and KIF11 were excluded) for CLEM (Fig. 4 and Fig. S5 a). The stage coordinates of the selected cells were saved and used to automatically guide the high-magnification imaging on the confocal microscope. For the liquid phase transfection experiment (Fig. 5), where only one phenotype was studied (COPB1), a total of 25 cells were selected, using only the property of diffuseness (Fig. 5 c) and selecting values higher than the average value of the control.

### High-magnification imaging light microscopy

For each of the cells selected by image analysis, the following automated scan job pattern was triggered: (a) cell coordinates were passed to the microscope and the stage was positioned such that the selected cell was centered on the optical axis, (b) software autofocus on DAPI signal of the target cell using 40× objective (NA = 0.75, Leica HCX APO U-V-I), (c) high-magnification z-stack acquisition (nine slices, 10.1 µm range) with 40× objective, 512 × 512 pixels, zoom 5, FOV 77.5 × 77.5 µm, channels 405 nm/488 nm, (d) imaging of the spatial context of the cell including the etched coordinate system with the 10× objective, 1,024 × 1,024 pixel, zoom 1.2, FOV 1.29 × 1.29 mm, channels 405 nm/488 nm/594 nm fluorescence, transmitted/reflected light. Communication with the microscope software was implemented in python using a library of functions that communicate with the Leica Matrix Screener software through the Leica CAM protocol network (Tischer et al., 2014; Tosi et al., 2021). Two functions were used, one to move to the specific coordinate calculated in the previous step and another to execute the acquisition of the described sequence of images, previously programmed using Leica LAS AF software (version 1.0.4, 2013).

### Correlation in light microscopy

CLEMSite is a set of software tools developed in python and C# to support automated correlative light and electron microscopy (CLEM; Fig. 1). The first of these tools, the CLEMSite-LM, was used to process the light microscopy images and extract landmarks from them (Fig. 2). The user provides a folder containing at least one image with two channels, one fluorescent with the GFP tagged organelle of interest and one showing the patterned glass-bottom grid. Both images were acquired simultaneously at low magnification with a FOV that included a full square and patterned letter inside, usually 600 µm². The grid pattern was acquired using reflected light (RL), the modality of the confocal microscope. Images had a pixel size of 1.7 µm/pixel with a dimension of 1,024 × 1,024 pixels.

A script was created to rename the folder images to a more readable format and then, the set of folders was given to the application CLEMSite-LM. This application reads the RL image and applies our algorithm LOD (Line Orientation Detector; Fig. S1). LOD applied a series of preprocessing steps in which gradients are selected positively if they follow a line. Pixel orientations were weighted with neighboring pixels by convolution with line morphological operators for each possible angle orientation. A projection of the image onto one axis from 0 to 180°, at a resolution of 1°, generated a 2D map where the main trend of a line inside the image could be detected by finding maxima using non-maxima suppression. Once the most prominent lines were found, iterative refinement steps were applied to logically discard the lines which are not likely to belong to the grid, e.g., sets of lines not crossing orthogonally or not keeping approximately the expected measures given by the manufacturer of the glass bottom dishes.

Points resulting from calculating line intersections on the grid were associated with their corresponding alphanumeric identifier and used as landmarks. The LOD parameters are dependent only on resolution, where the number of neighbors was set to (k = 12) and stroke size (stroke = 15) for 1,024 × 1,024 images. The parameters used for LM were between 0.06 and 0.075 for the Canny threshold, and a Laplacian filter was applied before in the presence of regular interference patterns or local contrast enhancement (CLAHE) when the brightness and contrast were unbalanced. The other provided parameters are the dimensions of the grid. In our case, we used MatTek grids, where the lattice is formed by a sequence of two lines spaced 40 µm followed by another space of 580 µm. We used as a landmark the center position of the small square formed by the intersection between two sets of perpendicular lines spaced 40 µm.

Each LM position in a map also requires a unique identifier. For landmarks, we conveniently named them after the two-character combination inside the nearest grid square (e.g., 4N). By convention, given a grid square with the inner pattern straightly oriented, the top left corner landmark is assigned (Fig. 2 b). To automatically identify the characters on each grid square we trained a U-net using a mixture of real (20%, 1,115 images) and synthetic (80%) binary images (128 × 128) of the identifier patterns combined with augmentation. The CNN architecture, implemented with Tensorflow (https://www.tensorflow.org/), used 6 convolutions layers in a sequential manner followed by two dense layers. The loss function used was categorical cross-entropy that converged after 70 epochs. The prediction of the network was additionally validated using previously detected neighbors and the expected position of the landmark (e.g., 1A can be a neighbor of 1B, but not of 8B).

Detected landmarks are saved by the application in image coordinates. Since the stage coordinates of the optical microscope (saved in the metadata of the image) refer to the center of

the image, the translation from pixel coordinates to stage coordinates can be obtained by simple addition after converting pixels to distances using the known image pixel size. Similarly, for each targeted cell, the difference in micrometers from the center of the image to the centroid of the object of interest was provided and converted to its respective stage coordinates.

## Electron microscopy

### Electron microscopy sample processing

The entire EM processing was performed using a Ted Pella Biowave Pro microwave. After samples were lightly fixed and imaged in the confocal microscope, they were heavily fixed with 2.5% glutaraldehyde (EMS), 4% formaldehyde (EMS) and 0.05% malachite green (Sigma-Aldrich) in 0.1 M PHEM (pH 6.9: 240 mM PIPES [Sigma-Aldrich], 100 mM Hepes [Biomol], 8 mM $MgCl_2$ [Merck], 40 mM EGTA [Sigma-Aldrich]). The samples were then postfixed with 0.8% $K_3Fe(CN)_6$ (Merck), 1% $OsO_4$ (Serva) in 0.1 M PHEM. The samples were stained successively with 1% tannic acid (EMS) and 1% uranyl acetate (Serva) to enhance the contrast. Samples were dehydrated in a graded ethanol series (30, 50, 75, 90, 2 × 100%) and infiltrated in a graded series of Durcupan (25, 50, 75, 90, 2 × 100%, Sigma-Aldrich) and polymerized in the oven at 60°C for 96 h.

### Correlation in electron microscopy

The central disk of the MatTek dish was broken out using heat shock. The resin disk, containing the cells along with the imprint of the coordinate system on the surface, was mounted on SEM stubs (Agar Scientific) with a conductive carbon sticker (Plano). To reduce the amount of charging the samples were surrounded by silver paint (Ted Pella, Inc.) and coated with gold for 180 s at 30 mA in a sputter coater (Q150RS; Quorum). The samples were introduced into the Crossbeam 540 (Zeiss) and positioned at 54°. CLEM*Site* is interfacing ZEISS Atlas 5 version 5.2.0.150 from Fibics Incorporated to navigate to the correct positions and to prepare the ROI for imaging. When scanning the surface of the sample with a scanning electron microscope (SEM), after detaching the glass from the resin, the imprinted grid pattern from the glass bottom dish and letter combination could be distinctly observed.

To optimize the visualization of the gridded pattern, samples are rotated using the FIB-SEM stage to orient the grid at a 45° angle with respect to the SEM image (Fig. 2 c). This ensures that both perpendicular orientations of the grid are efficiently detected when recording the secondary electrons which are best suited to visualize topological information. Cell contours could be visualized at higher accelerating voltages (5–10 kV) allowing signals to be detected from deeper regions inside the sample. However, cell visibility was sample dependent, and individual cells could only be differentiated one from another at lower cell confluency.

Once all LM images were processed and the landmarks and targets stored in their respective files, we proceeded with the FIB-SEM acquisition. Our software connected to the microscope control in a client-server architecture, where the client streamed information and commands which were parsed, validated, and then executed by a server. The server software relied on a dynamic library in NET provided by Fibics Incorporated to control the microscope through ZEISS Atlas 5. When the microscope was ready, a first image of the surface was acquired (1.5 kV, 700 pA) using the secondary electron detector (SESI) and sent to our client application. The reliability of the computational process was increased by having the user move to a grid square and indicate the combination of visible characters in that square. After the first image was computed, landmark references were calculated and mapped to an ideal coordinate system layout based on manufacturer measures (MatTek dishes: 560 × 560 μm, a line width of 40 μm for SEM), and initializing a linear system to predict further positions within the grid. Afterward, the stage was moved to the approximate position of each grid square close to the regions of interest to be acquired.

When applied to SEM images, the LOD algorithm (used previously in LM) had a high failure rate (from 0.05 to 5%).

In SEM images, grid lines are very often blurry or erased. Neural networks have proved to be very resilient to noise in classification and object detection (Ravindran, 2022). Based on those successes, we trained a U-Net network (Ronneberger et al., 2015) to provide the probability mask where edges of a crossing could be found (Fig. 2 c). Enough training data to optimize the network was provided with data from the LOD and used with the errors curated manually. Manual segmentation was performed using the corner shadows in around 100 difficult cases when LOD failed. We extracted a total of 600 images and augmented them to 3,000 images by variations in scaling, rotation, translation, and intensity values. As preprocessing steps after augmentation, CLAHE (32 × 32 filter size), Gaussian blur (Sigma-Aldrich 1, 5 × 5 filter size), and normalization were applied. Processed images were then used to train a convolutional autoencoder based on U-Net, using binary cross-entropy as a loss function, together with an Adam optimizer at a fixed learning rate of 1e-4. The network computed a probability map of the regions in the image that contained an edge belonging to a grid line. The last part of the previous LOD algorithm was adapted to find the peaks based on the maximum probability of lines and provided results in the form of image coordinates. Using the detection system based on a CNN the rate of failure in detection was reduced, with an average RMSE of detection originally of 12.66 ± 18.8 μm to one of 6.23 ± 6 μm with respect to ground truth (with n = 149), a considerable improvement. The implementation of all the networks was carried out with Keras 2.0.8 and Tensorflow 1.3.

The described detection step was performed for 30% of the grid corners, with a minimum of 20 grid corners required. The scanning of the surface sample (duration between 30–40 min) provided enough positions to generate a global registration with a targeting accuracy of 5–20 μm throughout a surface area of 1 cm². Since not all positions acquired in LM could be present in the current resin block (on several occasions the resin block broke into two different pieces), another CNN was trained to detect positions very close to the border or outside the sample, with a binary output (image valid or not). After the scan was finished, the positions of targeted cells were predicted using a global affine transformation, which uses all available matched points between LM and SEM. All the information from mapping

was stored in memory using a pandas dataframe for further usage and queries. If cells were closer than 150 µm one of them was removed because the trench and acquisition of the first cell could interfere with the acquisition of the adjacent second cell.

The generated map between LM and SEM was used before each acquisition of a targeted position. First, it was used to move to the target cell and calculate the coincidence point there. After this, any landmark in a radius of 1,200 µm close to the region of interest was imaged (usually resulting in four to eight neighboring landmarks) and stage coordinates were obtained again. Subsequently, for each predicted position accuracy of 2–5 µm was achieved. This was precise enough to hit the organelle of interest.

In the current workflow, analyzing the images from light microscopy to extract the positions of the cells and the glass bottom grid took ~2 h. This can be carried out on any computer at any time between the LM and EM session. In this step, extra verification steps were added that helped the researcher validate the current cells selected in the light microscopy images. The corresponding map of the grid in the FIB-SEM in the resin block is acquired in ~3 h. The initial setup for the first volume acquisition takes around 30 min, with minimal user input (brightness and contrast of detectors). After the first cell acquisition starts successfully, the microscope can run autonomously until the FIB-Gallium source has reached its limits and needs to be reheated.

### Automation of the FIB-SEM acquisition

To align the electron beam and the ion beam, we used an automatic coincidence point procedure to match both (Fig. S3 a). First, a square was burned onto the surface of the sample using the FIB. The geometrical relation between the two beams was used to move the stage to the coincidence point resulting in both beams hitting the same spot on the sample surface (Fig. 3 a). After this, a trench was milled in front of the ROI at a FIB current of 15 nA and a 20 µm depth. Polishing was performed at 3 nA. In this way, a cross-section through the ROI was exposed and the polygonal shape of the trench was detected (Fig. 3 b and Fig. S3 b). The field of view was positioned on the cross-section, and points with high variance were obtained to tune the automatic focus/stigmatism functions (Fig. 3 c). In the last step, the acquisition started with a crisp focus (Fig. S3 c) and the system acquired section after section (slice and view). For 3D data acquisition, the FIB was operated at 1.5 nA with the SEM and the FIB operating simultaneously (Narayan et al., 2014). The images were acquired in analytical mode (1.5 kV, 700 pA) using the energy-selective back-scattered electron (EsB) detector at 1,100 V grid voltage. The dwell time was set to 10 µs/pixel, with a line average of 1 and an 8 nm pixel size. The FOV was set to 25 × 15 µm (XY front face) and images were collected for a ROI of 25 × 30 µm (XY, plane parallel to the surface) at 200-nm intervals for COP phenotypes, resulting in 8 × 8 × 200–nm voxels [XYZ]). Note that to gain time in the preparation process for a run, we have not covered the ROI with a platinum protective layer and alternatively we increased the thickness of the gold coating of the full sample. In such cases, only low z-resolution acquisition is possible, as acquiring at a higher resolution would require sputtering of the sample surface.

During the acquisition, an additional process was launched to monitor the status of the imaging (Fig. S3 b). This monitoring was in charge of automatically placing the region where the autofocus and autostigmatism (AFAS) routines had to be executed. The AFAS routines happened at periodic 45 min intervals and were executed in a region of the cell where high contrast could be found. This was achieved using a Harris corner detector (Harris and Stephens, 1988) combined with clustering: the clustering with a higher number of corners is the candidate point for AFAS.

In addition, the imaged x-y region of interest was tracked in the y-direction to prevent undesirable drifting. Cross-correlation was employed to find the relative difference between consecutive sections. The upper layer of the resin covered with gold was detected using changes in entropy and a drift >25% of the total y size, which was corrected at 0.5 µm increments, maintaining the region of interest in the desired field of view. After the acquisition of one position was completed, the stage of the FIB-SEM was moved to the next ROI and started a new acquisition.

### Stereology

All stereological measurements were performed using IMOD (Kremer et al., 1996). From every siRNA treatment, a minimum of two cells typical for the individual treatment and very different from the negative control were selected by the image analysis pipeline. The FIB-SEM images were acquired throughout the cell of interest with a spacing of 200 nm and a random starting point producing 10–20 evenly spaced sections per cell. Golgi cisternae were defined as membranous structures devoid of ribosomes with a threefold length to breadth ratio. Golgi stack profiles were defined as any assembly of cisternal membranes and the total area enclosed by them (Lucocq et al., 1989). The volume of the Golgi stack, (V [Golgi stack]; Fig. S4 c) was estimated using a point counting-based Cavalieri estimator by applying a regular point lattice that yielded 100–200 point hits over Golgi stack profiles per typical section stack (Lucocq et al., 1989).

Our FIB-SEM sections were prepared orthogonal to the monolayer substratum in a haphazard (random) orientation relative to the analyzed cell, and therefore comprise vertical sections on which the surface density of the Golgi cisternae in stack volume can be estimated by counting intersections (I) arrays of cycloid line probes (Ferguson et al., 2017). As cisternal membrane morphology was often indistinct, Golgi cisternae profiles were defined as a single line bisecting the cisternal membrane profile. The mean number of cisterna was estimated using the ratio of (intersections with Golgi cisterna; I[cist]) to (intersections with the Golgi stack profile face; I [stack face]).

## Software availability

CLEMSite software can be downloaded at https://github.com/josemiserra/CLEMSite. Supplementary instructions and

## Online supplemental material

Fig. S1 shows a diagram explaining the algorithm for line detection and landmark recognition in LM and SEM. Fig. S2 shows examples of successful and failed landmark detection on SEM images (SE detector) from surfaces of different samples. Fig. S3 shows diagrams depicting the algorithm steps for (a) automatic coincidence point detection, (b) milling and trench detection and (c) automated AFAS steps for the autonomous workflow. Fig. S4 shows in (a) the python CLEMSite-EM user interface and in (b) a diagram depicting the Run Checker steps during the acquisition. Fig. S5 shows the phenotype description and stereological quantification of chosen cells for the entire workflow. Video 1 is related to Fig. 5 and shows the automatically acquired FIB-SEM cell volumes for the COPB1 experiment (Fig. 5). Video 2 is related to Fig. 5 and shows the second part of automatically acquired FIB-SEM datasets (Fig. 5). Table S1 showing the RMSE error for global and local targeting position in four experiments. Table S2 shows the description of the 32 siRNA spots added to one dish in Case Study 1 (Fig. 4). Table S3 shows the number of failures in targeting cells for the different experiments. Table S4 shows the different types of failures detailed per experiment.

### Data availability
The datasets from Fig. 5 a (COPB1 knockdown), also shown in the supplementary video are available in the EMPIAR repository (Iudin et al., 2016; accession number EMPIAR-11314).

## Acknowledgments
We thank the staff of the Electron Microscopy Core Facility and the staff of the Advanced Light Microscopy Facility at EMBL, Heidelberg. We thank Alexandre Laquerre and Ken Lagarec (Fibics Incorporated, Canada) for all the assistance in ZEISS Atlas 5 software and Carl Zeiss Microscopy GmbH for support and feedback. We thank the Electron Microscopy Core Unit at the Max-Planck-Institute of Experimental Medicine Göttingen for access to their Crossbeam.

This work was supported by EMBL funds and by by the Deutsche Forschungsgemeinschaft (DFG, German Research Foundation) – Project number 240245660 – SFB 1129 (project Z2).

CLEMSite is an open source software developed to control the operations of a microscope produced and distributed by Carl Zeiss Microscopy GmbH and interfaces with a software (ZEISS Atlas 5) developed by the company Fibics Incorporated. At the time of the realisation of this work, M. Holtstrom., D. Unrau., were employed by Fibics Incorporated and R. Kirmse was employed by Carl Zeiss Microscopy.

Author contributions: J.M. Serra Lleti: Conceptualization, Data curation, Formal analysis, Investigation, Methodology, Software, Validation, Visualization, Writing-original draft, Writing-review editing. A.M. Steyer: Conceptualization, Formal analysis, Investigation, Methodology, Validation, Visualization, Writing-original draft, Writing-review editing. N.L. Schieber: Conceptualization, Investigation, Methodology, Validation, Visualization, Writing-original draft, Writing-review editing. B. Neumann: Methodology, Validation, Writing-original draft, Writing-review editing. C. Tischer: Methodology, Software, Validation, Writing-original draft, Writing-review editing. V. Hilsenstein: Methodology, Software, Validation, Writing-original draft, Writing-review editing. M. Holtstrom: Methodology, Software, Validation, Writing-original draft, Writing-review editing. D. Unrau: Methodology, Software, Validation, Writing-original draft, Writing-review editing. R. Kirmse: Methodology, Software, Validation, Writing-original draft, Writing-review editing. J.M. Lucocq: Methodology, Validation, Writing-original draft, Writing-review editing. R. Pepperkok: Conceptualization, Resources, Supervision, Writing-original draft, Writing-review editing. Y. Schwab: Conceptualization, Funding acquisition, Resources, Supervision, Writing-original draft, Writing-review editing.

Disclosures: J.M Serra Lleti reported non-financial support from Zeiss Microscopy during the conduct of the study. "Since the developed work required control of the microscope hardware used for the experiments described in the manuscript, there was an agreement with Zeiss, that Fibics Incorporated, a third-party company from Zeiss, would provide a developer's API able to command certain aspects of the microscope (details are described in the manuscript). The provided developer's API was experimental and required technical support from Fibics Incorporated side. There were also meetings to discuss technical details about the operation of the microscope in the context of the project, where we benefited from the company's expertise in FIB-SEM technology. For this reason, the API developers that provided such support are cited as co-authors (David Unrau and Mike Holtstrom)." No other disclosures were reported.

Submitted: 4 October 2022

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

# Supplemental material

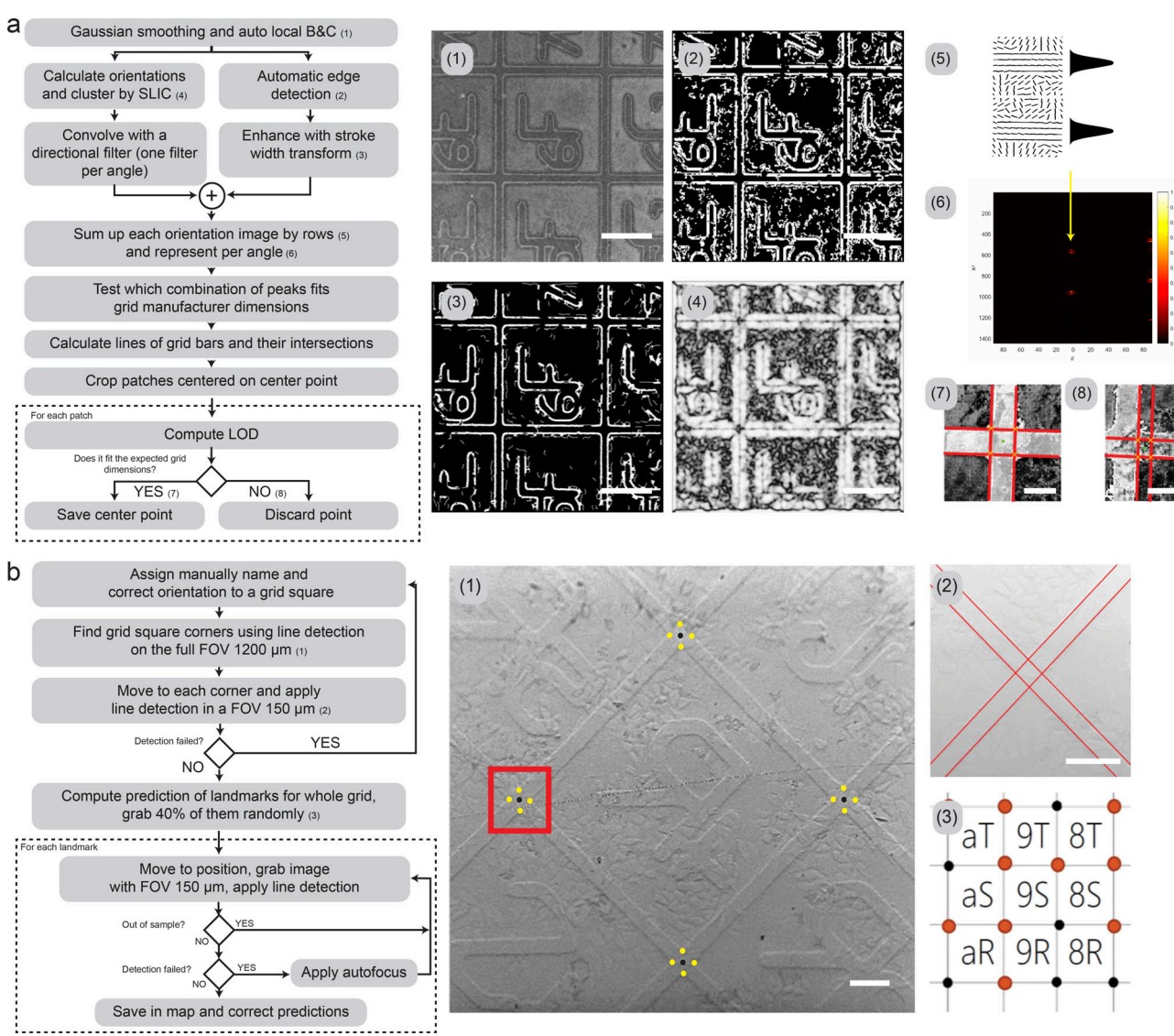

Figure S1. **Line detection and landmark recognition in LM and SEM. (a)** Schematic of the line detection algorithm. Each step is illustrated with the corresponding output image: (1) Reflected light image (LM) of the coordinate system is smoothed and the brightness and contrast are automatically balanced with adaptive histogram equalization. (2) Automatic edge detection is performed using Canny edge detection and non-maxima suppression (NMS). (3) Image edges are enhanced with stroke width transform, which analyzes all gradients to keep only the ones belonging to the imprinted grid. Thus, the image is cleaned to facilitate the recognition of the alphanumerical pattern. (4) Pixel gradient orientations (from Sobel operators) are extracted and homogenized in superpixels (SLIC algorithm), where similar orientations get clustered to the same superpixel. (5) The image resulting from 4 is convolved by every angle from 0 to 180°, and all the rows of the image are added to form a vector projection. Vectors are arranged in a matrix from 0 to 180. (6) From 5, peaks are found using non-maxima suppression and the repetition and spacing pattern are tested to find the best fit to the grid dimensions according to the manufacturer. Each peak is the result of a line detected in the image and in this way it can be plotted back in the original image. With the line detected, by calculating all the intersections between lines, the grid bar crossings can be found. (7) For each bar crossing, a refinement is applied. First, the area surrounding the crossing area is cropped, and the patch is analyzed again (same line detection algorithm) to validate the previous detection of the lines. When the distance between intersections is not fitting the expected separations of the grid pattern (i.e., 20 μm thickness for the border and 580 μm for the square with the alphanumeric pattern, with some additional tolerance), the landmark is not accepted. (8) This might happen when dirt or scratches make the detection algorithm fail. The final result of this process is, first the list of references based on the detected central positions of the grid bar crossings (landmarks), and the cropped character (as shown in Fig. 2 b). The cropped character is passed to a convolutional neural network, and the alphanumeric character is automatically identified (for details about this, see supplementary materials notebooks 1 and 2: https://github.com/josemiserra/CLEMSite_notebooks). Each landmark is then renamed based on the corresponding detected character. **(b)** Schematic of the algorithm used by the *Navigator* module to find landmarks in the SEM, to build a map based on the grid. (1) In the first step, the SEM is positioned at a random square in the MatTek grid. The software detects the corners (black dots) by detecting the line intersections of the square edges (yellow points). The process is the same as the one explained in (a). (2) Each corner is refined by applying the line detection (red lines) in a higher magnification view. To optimize the process and reduce the amount of SEM images of the sample surface, the detection procedure is applied to only a group of randomly selected landmarks in the MatTek grid. (3) By getting a 40% of total landmarks, and sampling them with a uniform random distribution, it can be achieved with similar accuracy as when scanning the full dish. If the line detection fails, autofocus is applied once. If after a second round, the detection fails, the landmark position is flagged as blocked. In this way, landmark positions that fall outside the sample or are too damaged, are discarded from the final landmark map. Once a new position is saved in the map, if it is considered a valid position (not blocked), then local and global transformations are recomputed and updated. Scale bars: (a; 1–4) 200 μm; (7, 8) 25 μm; (b; 1) 100 μm, (2) 50 μm.

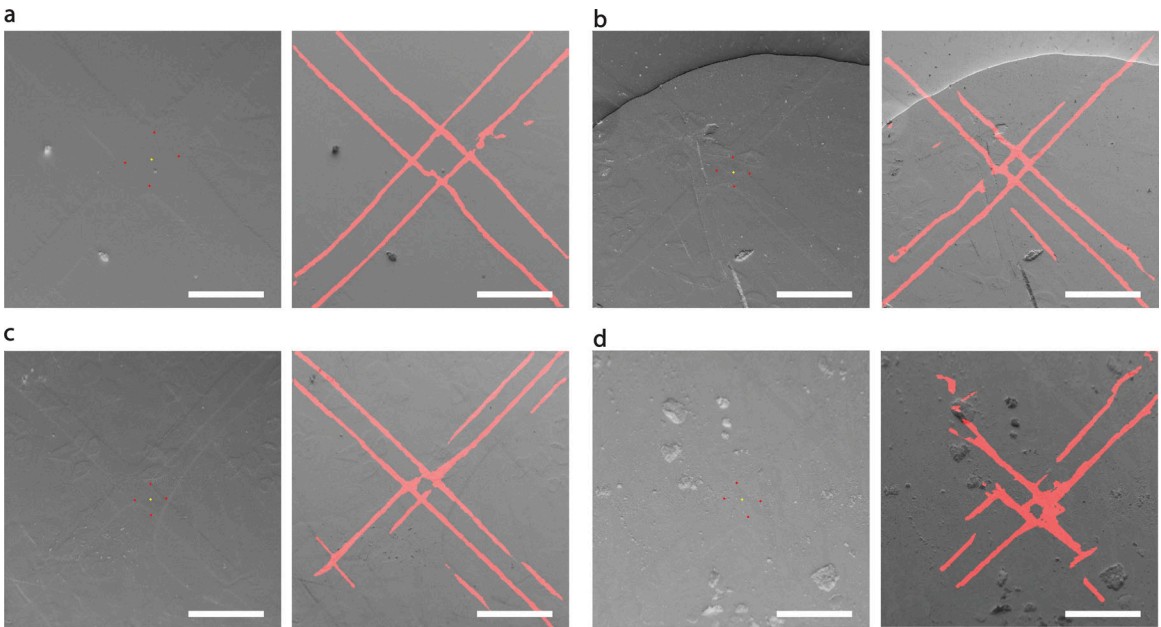

Figure S2. **Examples of landmark detection on SEM images (SE detector) from surfaces of different samples.** Cracks, scratches, and dirt on the surface make landmark detection difficult and more error-prone. For each square, the left image shows the final detection, with the yellow dot representing the detected center position of the crossing and the red points the corners of the crossing. The right image is the same with inverted brightness and contrast, with red pixels representing the probability of being a grid edge as detected by the neural network. The probability map from the neural network is the result of the network inference, with the set of images used during training different from the images used as input during the experiment, which is shown here. We observe that the neural network can generalize very well the detection of the grid patterns in the resin surface. Here we exemplify the common cases that can lead to an error in the detection of a landmark. **(a)** The sample is in a perfect state. **(b)** A crack present in the upper part might affect the predicted accuracy of the overall map, even if the detection is identified as good (or close to it). **(c)** Scratches can be the cause of false positives for the grid detection, in this case, scratches parallel to the grid bar. Even if this specific error was later corrected by taking also into account the length of the line stroke, we presume that longer scratches than the ones shown in the exemplary image could cause the same problem again. **(d)** In other cases, dirt and other material residues, e.g., from silver painting (used around the sample border to derive charges), might mislead the detection algorithm and increase the final error. The detection problems might change on a sample basis. A detailed analysis of the error detection is shown in the supplementary material in notebook 2 (https://github.com/josemiserra/CLEMSite_notebooks). Scale bars: all 100 µm.

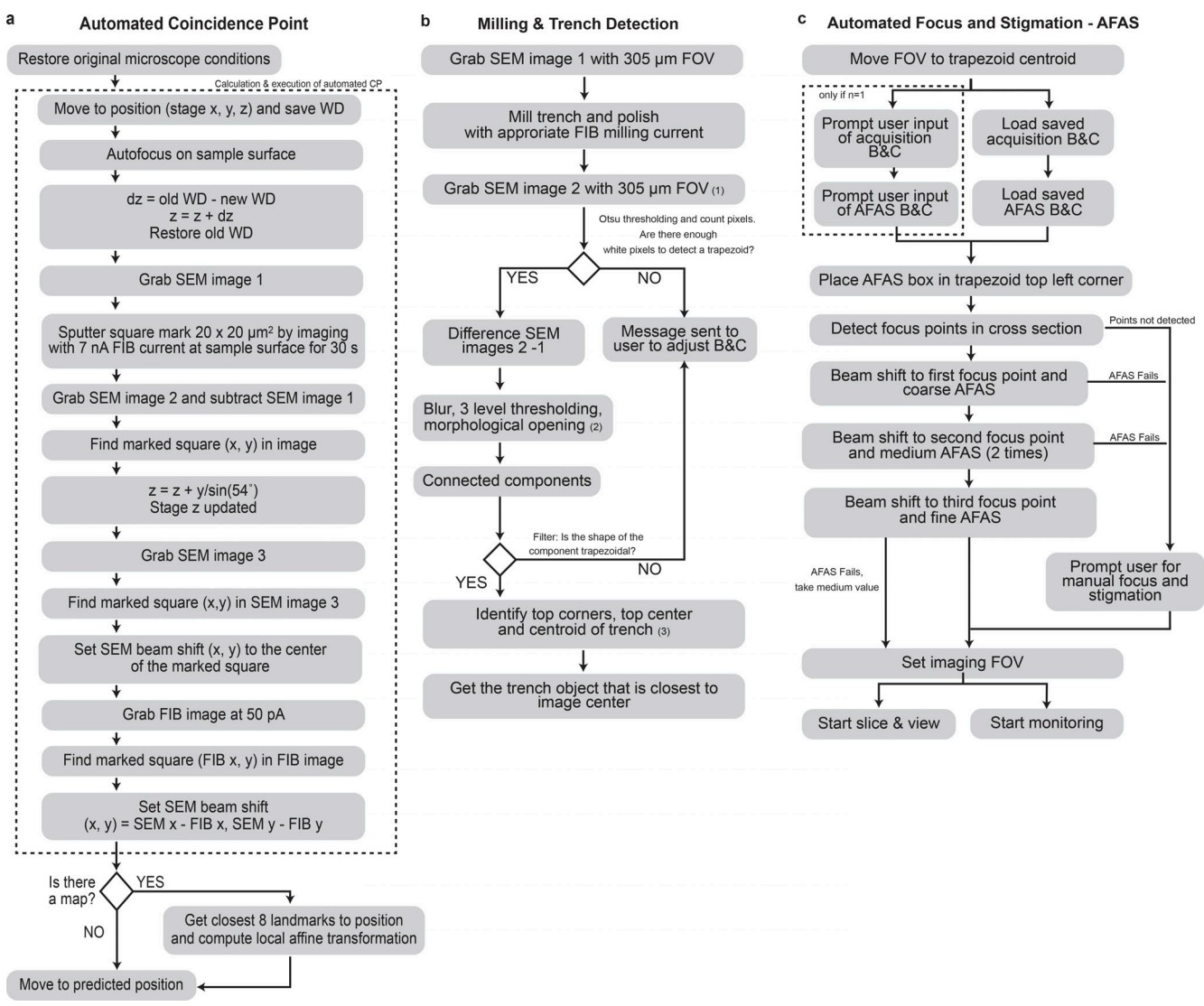

**Figure S3.** **Automatic workflow setup for data acquisition in the FIB-SEM (*Multisite*). (a)** Flow diagram of the algorithm used before each target cell is acquired. The boxed part (dotted line) indicates instructions belonging to the coincidence point (CP) calculation. "WD" refers to Working Distance (distance to the focused object on the z-axis). "Grab" refers to commanding the microscope to acquire an image of the surface of the sample. (x, y) indicates that the action takes place in respective stage coordinates in the x, y-axis. "dz" is the difference in z position, SEM x, SEM y—stage position coordinates x and y using the SEM detector. FIB x, FIB y stage position coordinates x and y using the FIB detector. In both cases, pixel coordinates from the image are translated to stage position coordinates given by the center position of the image. Upon completion, when a stored map of landmarks is present (there are surrounding grid bar crossings to the cell target), the closest 8 landmarks are used to compute a local transformation that will re-estimate the cell position with higher accuracy. **(b)** Flow diagram of the algorithm used for Milling & Trench Detection. Numbers (1), (2), and (3) correspond with images (1), (2), and (3) in Fig. 3 b. After the trench is milled, a quick routine examines if the B&C (brightness and contrast) is good enough to differentiate the trench from the background. If not, the user is prompted to adjust the B&C until the trench is visible. Since simple thresholding is usually not enough, the detection of the trench is repeated on the new image using a three-level thresholding algorithm after a slight blur. This algorithm is fast and identifies and groups pixels as belonging to three categories. The darkest category is usually the trench. The thresholded object is then identified if its geometry has a trapezoidal shape, to differentiate it from other con-founding objects. If several trapezoids are present (from previous acquisitions), the closest to the center is taken as a reference. In the trapezoid, the top center position can be used as a reference to focus the FOV (field of view). **(c)** Flowchart of the routine used for setting the conditions before the acquisition, after (b). In the automation routine, the user must decide the brightness and contrast (B&C) of the sample only for the first cell acquired (*n* = 1). Values of B&C will be stored for future acquisitions. After choosing an optimal B&C, the goal is to start with a crisp image with a good focus and stigmatism set of values. The core *AFAS* routine is provided by ZEISS Atlas 5 software and is triggered in a reduced window from the full field of view (FOV) at different magnifications, from lower to higher. At each magnification, high complexity regions are found to be the center of the window where the *AFAS* is applied. If this routine fails to find a good focus before starting to acquire, which could happen in exceptionally damaged samples, the user is prompted to focus manually and the values of focus are taken as reference for the next acquisition.

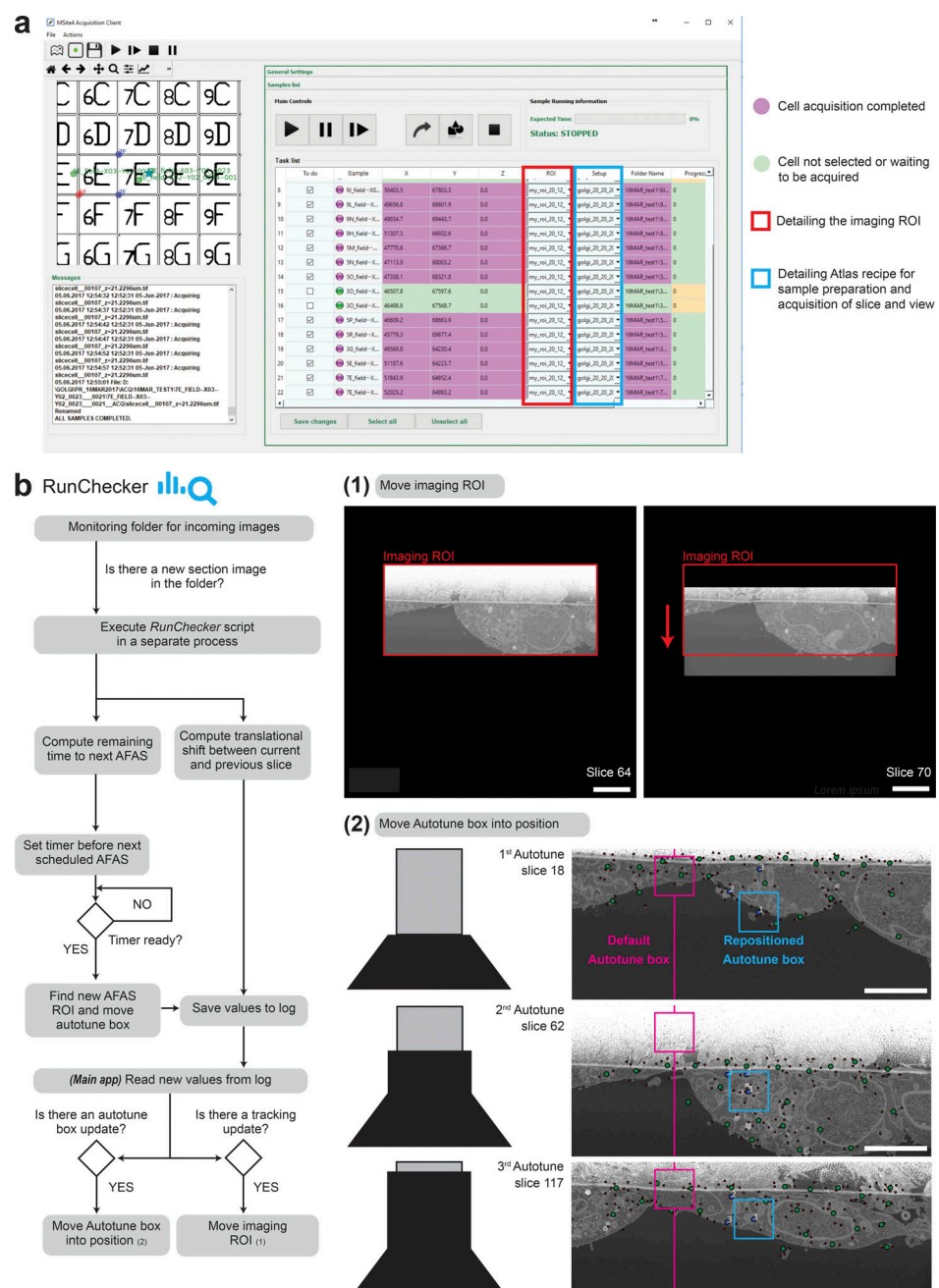

Figure S4. **CLEM*Site*-EM interface and *Run Checker* details. (a)** Screenshot of the CLEM*Site*-EM interface to outline the details of the software User Interface (UI). In the top left panel, a map depicts targets (green) and landmarks (blue if SEM stage coordinates are matched with light microscopy stage coordinates, red if no match is present). Bottom left: A messaging console is used to display the communications with the server and which instructions are sent to the microscope. The right panel displays the list of all the targets to be acquired. The list presents which targets are already acquired (purple) and which ones are intact (green). Targets can be selected or deselected by ticking the "*To Do*" checkbox in the first column. For each target, it is possible to decide on a rectangular field of view of the cross-section in x and y and assign it here to each phenotype according to its expected size (ROI, red outline). In the last modifiable column, the ZEISS Atlas 5 recipes for the actual acquisition, which includes the size of the section imaged from the total 3D volume milled by the FIB-SEM (Setup, blue outline). The last two columns show the individual folder where the acquisition is saved and the percentage of progression during the acquisition. **(b)** Flowchart of the logic applied by the *Run Checker* module. This module becomes active once a run starts and triggers a script each time a newly acquired image is stored in the folder. During the progression of the acquisition, the FOV carries a translational shift that has to be tracked and corrected continuously. In this module, a routine calculates the translation between two consecutive frames, and given the incremental shift, it decides to move the imaging ROI if the sample has drifted with respect to the image acquired at the beginning of the acquisition (1). The reference used to track is the upper coating, which cannot be drifted more than a tolerance (one-fourth of the image height). If that happens, the FOV is moved up or down respectively. The same principle is applied to the position of the autotune box (small window where the *AFAS* is applied, magenta and blue squares) which is moved into a new position before a new *AFAS* is executed (2). In this case, the image is analyzed to find optimal positions for the autotune box, first executing the same algorithm as used in Fig. 3 c, but now with the hard constraint that the position must be in the upper part of the image (half of the image height) and below the upper coating. The image coordinates are translated to FOV coordinates and the autotune box is repositioned.

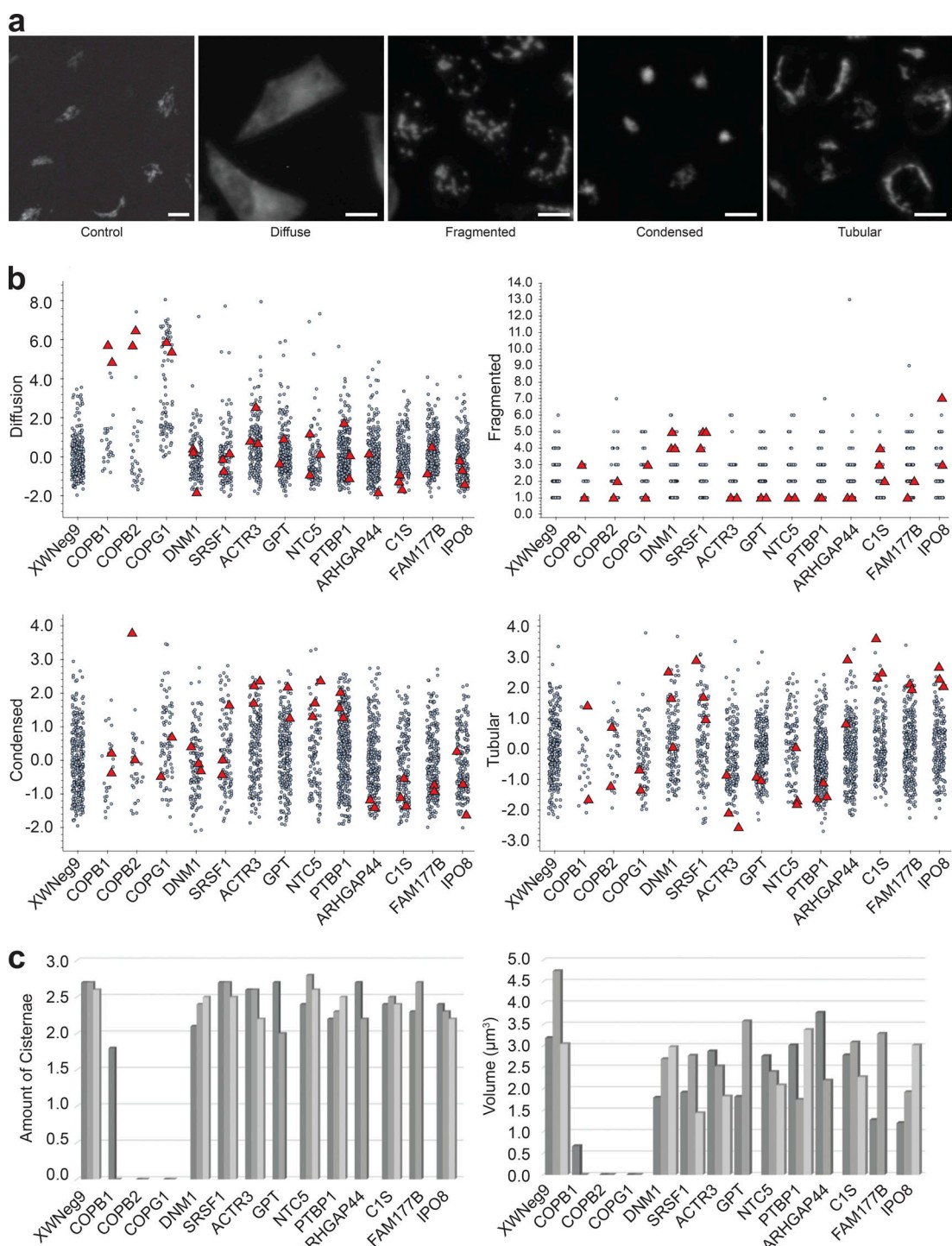

Figure S5. **Phenotype description and stereological quantification of chosen cells for the entire workflow. (a)** Illustrations of the different Golgi phenotypes revealed by the GalNac-T2-GFP signal: control, diffuse (COPG1), fragmented (DNM1), condensed (ACTR3), and tubular (IPO8). Scale bars: control, 5 µm, rest, 10 µm. **(b)** Scatter plots of computed features measuring the strength for each phenotype. Each gray dot represents the feature value associated with one cell normalized respect to the mean. The x-axis displays the corresponding siRNA treatment (ACTR3 $n = 183$, ARHGAP44 $n = 282$, C1S $n = 179$, COPB1 $n = 26$, COPB2 $n = 34$, COPG1 $n = 88$, DNM1 $n = 137$, FAM177B $n = 252$, GPT $n = 260$, IPO8 $n = 194$, NT5C $n = 115$, XWNeg9 $n = 305$, PTBP1 $n = 357$, SRSF1 $n = 115$). Diffuseness, condensation, and tubularity values are normalized with respect to the control (Neg9). Fragmentation illustrates the number of fragments detected in the Golgi apparatus. Red triangles highlight each one of the selected cells for the CLEM experiment (a total of 33). **(c)** Stereological quantification was applied on FIB-SEM images of the corresponding cells to measure the number of cisternae (left) and the volume (right) of the Golgi apparatus. Each bar represents the value measured for one cell, grouped by siRNA treatment. Since the sample size is very small ($n = 2$ or $n = 3$ per treatment), the screen was oriented exclusively to find large effects. Knockdowns of the COP proteins (COPB1, COPB2, COPG1), revealed a disappearance of the Golgi stacks (thus, no cisternal volume can be measured) replaced by a large accumulation of small vesicles. No obvious morphological differences were found in other siRNA treatments with respect to the control cells.

Video 1.   Fig. 5 **part 1.** FIB-SEM datasets of the automatically acquired data for the 25 cells (GalNAc-T2-GFP) of the COPB1 knockdown (48 h after liquid phase transfection knockdown) shown in Fig. 5. The data were acquired with a Crossbeam 540 (Zeiss) at a voxel size of 8 × 8 × 200 nm using CLEMSite. The images were inverted and aligned using SIFT in Fiji (Schindelin et al., 2012).

Video 2.   Fig. 5 **part 2.** FIB-SEM datasets (part 2) of the automatically acquired data for the 25 cells (GalNAc-T2-GFP) of the COPB1 knockdown (48 h after liquid phase transfection knockdown) shown in Fig. 5. The data were acquired with a Crossbeam 540 (Zeiss) at a voxel size of 8 × 8 × 200 nm using CLEMSite. The images were inverted and aligned using SIFT in Fiji (Schindelin et al., 2012).

**Provided online are Table S1, Table S2, Table S3, and Table S4. Table S1 lists of each mesenchymal cell used in the study. Table S2 the 32 siRNA spots used in the study. Table S3 shows number of volumes targeted used in the study. Table S4 shows number of failures and their cause used in the study.**

