## [Peer Review File · The Journal of Cell Biology]

CLEMSite, a software for automated phenotypic screens using light microscopy and FIB-SEM

José Serra Lleti, Anna Steyer, Nicole Schieber, Beate Neumann, Christian Tischer, Volker Hilsenstein, Michael Holtstrom, David Unrau, Robert Kirmse, John Lucocq, Rainer Pepperkok, and Yannick Schwab

Corresponding Author(s): Yannick Schwab, European Molecular Biology Laboratory

Review Timeline:

Submission Date:	2022-10-04
Editorial Decision:	2022-11-10
Revision Received:	2022-11-28

Monitoring Editor: Jodi Nunnari

Scientific Editor: Dan Simon

Transaction Report:

DOI: <https://doi.org/10.1083/jcb.202209127>

Revision 0

Review #1

1. Evidence, reproducibility and clarity:

Evidence, reproducibility and clarity (Required)

****Summary:****

Serra Lleti et al. report a new software (CLEMSite) for fully automated FIB-SEM imaging based on locations identified beforehand in LM. The authors have implemented routines for automatically identifying common reference patterns and an automated FIB-SEM quality control. This allows autonomous data acquisition of multiple locations distributed over the entire sample dish. CLEMSite has been developed as a powerful tool for fast and highly efficient screening of morphological variations.

****Major comments:****

The performance of CLEMSite has been demonstrated by the authors with two typical biological example applications. The stated performance parameters such as correlation precision and reproducibility are highly convincing and supported by the presented data. The authors give detailed information on their workflow and how on to use CLEMSite, which should allow other researchers to implement this for their own applications. The only comment I have in this regard, and I might have overlooked it, but how will CLEMSite be made available to the scientific community?

****Minor comments:****

The author mention that decreasing the z-resolution to 200 nm steps was critical to achieve high throughput. For applications that require higher resolution: is the only disadvantage a longer data acquisition time or are there also other limitations? I would assume that locating the finer structural details in a much larger data set might also introduce additional challenges in the data analysis pipeline.

In Table 1 in the supplements, the units are missing for the targeting positions.

On page 4, 4th line from the bottom, there is a typo in "reaaching a global targeting...".

2. Significance:

Significance (Required)

With CLEMsite, the authors present a powerful new software tool for the FIB-SEM imaging community. The high level of automation allows high throughput data acquisition with minimal user interaction. To my knowledge, this is the first software that fully automatically recognises reference features and is able to run fully autonomously after points of interest have been selected in FM. This high throughput screening tool for FIB-SEM imaging would make a substantial technical contribution to the field of cellular imaging.

My own expertise lies in the field of technical developments for CLEM and super-resolution FM. I am not able to judge the biological content of the manuscript.

3. How much time do you estimate the authors will need to complete the suggested revisions:

Estimated time to Complete Revisions (Required)

(Decision Recommendation)

Less than 1 month

Review #2

1. Evidence, reproducibility and clarity:

Evidence, reproducibility and clarity (Required)

Review on "CLEMsite, a software for automated phenotypic screens using light microscopy and FIB-SEM" by Serra Lleti et al.

The manuscript describes a toolset to correlate LM data with automated FIB-SEM of selected regions of interest. This allows 3D correlative microscopy of multiple adherent cells –from a single resin block. This allows much needed high throughput in CLEM analysis to become quantitative. Two applications on Golgi apparatus morphology are shown.

****Major questions:****

- The software has been developed in collaboration between Zeiss/ Fibics in collaboration with academic groups and will only function on Zeiss SEMs that have the proper software. Thus, if I understand correct, it will not be of generic use and a more appropriate title would be 'CLEMsite, a software for automated phenotypic screens using light microscopy and Zeiss FIB-SEM'
- How is the described approach using FIB-SEM advantageous compared to methods like Serial Block-face EM (SBEM) and array tomography using serial section where larger fields of multiple cells can be imaged? Especially because the axial resolution was set to 200 nm and

discussed as essential for the throughput speed.

- Is the data FAIR available?
- How is CLEMsite available? Is the code public or for sale?

****Other comments:****

- Can you comment on the flexibility of this method? It is described as a flexible method, but only HeLa cells (quite flat cells) and Golgi apparatus targeting was used. What about different cell types and what about targets with a less obvious EM morphology?
- For EM acquisition ZEISS smartSEM with ATLAS was used. LM was recorded with a microscope from a different vendor. Can the software be used regardless of microscope type?
- Create less variation in the size of scale bars.
- M&M: High resolution light microscopy: Why call this 'high resolution'? Specs given seem randomly chosen: For example objective magnification yes, NA not; excitation wavelength yes, emission not.

2. Significance:

Significance (Required)

See above: This depends on the availability of code, as well as the usability in FIB-SEM that is not based on Zeiss.

3. How much time do you estimate the authors will need to complete the suggested revisions:

Estimated time to Complete Revisions (Required)

(Decision Recommendation)

Cannot tell / Not applicable

Review #3

1. Evidence, reproducibility and clarity:

Evidence, reproducibility and clarity (Required)

****Summary:****

Schwab and coworkers present an automation software for correlative light and electron microscopy (CLEM) to acquire high-resolution SEM volume datasets at room temperature. This

automation enables large-scale data collection and morphometric analysis of cell phenotypes.

The paper is overall well written, but often assumes a lot of prior knowledge of the workflow, which might not be present in a general audience or for newcomers to the technique. This is also seen in the insufficient labeling and explanation of the figures. They seem a bit like presentation slides, which could be well understood with the help of the presenter/narrator, but alone lack a lot of information (see more specific comments below).

****Major Comments (in no particular order):****

- Final accuracy of $\sim 5 \mu\text{m}$... is this really sufficient? Given that the size of many mammalian cells is $\sim 10\text{-}15 \mu\text{m}$, this is still a HUGE error. Of course, there is a tradeoff between throughput and accuracy, the area covered and speed. Nonetheless, this means a serious limitation in terms of the kind of targets / biological questions that can be addressed with this technique! (Especially in the context of "rare events") This should be discussed in more detail.
- Given that the whole point of the paper is "large scale automation", I would have preferred a few more examples/higher n-count. A comment on which type of targets the authors envision/have validated would be nice (also in the context of the limitation in accuracy).
- It should be mentioned somewhere that "commercial dishes or coverslips" contain an imprinted grid pattern with numbers and letters to locate specific squares. [Again: probably clear to "aficionados" of the technique but totally unclear to newcomers/outsideers]
- "It is important to keep the initial number high in order to compensate for the loss of targets" - what % of targets is lost exactly in the final step (FIB-SEM imaging)? The 10 cells out of 35 (29%!) that were not "of sufficient quality for further downstream analysis", were they lost/discarded because of problems in the automation (e.g. autofocus/tracking failure) or for other reasons (e.g. preservation of the cells during fixation/embedding)?
- "One essential paradigm shift for increasing the acquisition throughput is the decision to decrease the resolution in the z-dimension, thus prioritizing the speed of acquisition and ultimately the total number of cells acquired in one run.". Surely, reducing z-resolution is an obvious way to speed up acquisition times. But this is not tied to the use of this software and obviously comes at a price ... this has been discussed before and is nothing novel. Hence "paradigm shift" might be a bit too strong. I however fully agree with CLEMSite's potential as a screening tool. Could a "high resolution" (isotropic) mode not be implemented, too? [then it would be up to the user to decide what to prioritize - throughput or resolution]
- There is no mentioning of why this specific hardware was used. Are there any limitations that currently restrict the approach to Zeiss machines? Any plans supporting other vendors? Of course, there are always certain benefits with certain instruments. Or just simply no others were available...
A comment on which part was performed by/at Zeiss and which in the labs would be useful to understand specific contributions. (Since a conflict of interest statement seems missing).

****Figures:****

Figures should be improved. They often contain too little information to understand the concepts/results discussed and there's lots of white space. The legends should be improved accordingly. In general, a more concise and structured figure design could go a long way of improving the quality of the manuscript. Please find a few suggestions (for the main figures) below (but the same should be applied to the supporting figures):

Fig. 1: While I believe it is clear to me what each scheme is supposed to represent, someone less immersed in this topic (or just entering the field) may have problems navigating the figure. For example: what are all the different letters and numbers? What's the blue box with the trapezoid ("EM targets" - it may become clear later, but here it is not), what are the blue and the red arrowhead, respectively (I suppose EM and focused ion beam?). This should be improved and labeled accordingly.

Fig. 2: Again, a lot of annotation is missing. E.g. what is the 3rd insert in b exactly (edge-detection? After CNN identification?)? For most of the figure, yellow squares are used to indicate the zoom-in region, why not for b) 1st row? With the "zoo" of scale bars, wouldn't it make sense to either always show the same bar (e.g. 200 μm), or scale the images so things become more comparable? In this regard: a) 2nd column and e) 1st column represent the same FOV. Why are they shown with different magnification/cropping?

Fig. 3: a) The procedure is well described in the legend, but no motivation is given in the text, why this is necessary. c) There's some floating density in the white space. Is this due to thresholding?

Fig. 4) Again, description/labeling of the figure is poor. E.g. what are the red outlines present in c) row 1-3 (but missing in row 4; why?)? [presumably these are the siRNA spots?] Is there any reason this figure could not be further sub-divided into d), e), f) etc? As it stands, a lot of additional descriptors ("second from the left", "two images on the right") are necessary while a simple call to a), b), c) would be much easier...

Fig. 5) Additional labeling (a,b,c...) could be helpful here, too. While intuitively I would assume that blue = DAPI and green = GFP, these things should be labeled or described in the legend. Especially in the 3D rendering it is quite unclear what is being portrayed. Is this an overlay of a FIB/SEM segmentation with the confocal 3D-data?

****Minor:****

The first element in the filtered list can thus be stored for the subsequent application of autofocus and autostigmation procedures (AFAS) (Supplementary Fig. 3c). [technically this has been defined before]

****Typos/grammar:****

In our case, we had two of such experiments, reaching a global targeting accuracy (RMSE) of 8

{plus minus} 5 μm .

A transformation is computed to register together the LM list and EM landmarks list, ...

The FOV is magnified from a 305 μm by 305 μm to a 36.4 μm by 36.4 μm surface area and an image of the cross-section is taken (Fig. 3c).

The sample is positioned at the target coordinates of the first cell, and the Multisite module performs the coincidence point alignment of both the electron and ion beams (Fig. 3a and Supplementary Fig. 3a).

To preserve the target, the sample is drifted 50 μm in x. [shifted?]

2. Significance:

Significance (Required)

****Significance:****

It is clear, that the kind of automation outlined here is necessary to elevate correlative SEM volume imaging to a "high-throughput" technique, which could become valuable for many biological questions. CLEMsite offers a valid technical solution and appears to be a solid implementation of the traditional/manual workflow. However, its presentation needs to be improved before we can support publication.

3. How much time do you estimate the authors will need to complete the suggested revisions:

Estimated time to Complete Revisions (Required)

(Decision Recommendation)

Less than 1 month

Manuscript number: RC-2021-00990R

Corresponding author(s): Yannick Schwab, Anna M Steyer

1. General Statements

We are grateful to the reviewers for their time and expertise, and we have addressed all points they raised as detailed in our point-to-point response and highlighted the changes in the main manuscript. We have addressed all points raised by the reviewers and elaborated how this was done in a point-by-point reply. There are two new tables and a new supplementary figure. The figures and the text have been reshaped, according to the suggestions.

We are looking forward to your reply.

Best regards, Yannick Schwab and Anna M Steyer

Reviewer #1 (Evidence, reproducibility and clarity (Required)):

****Summary:****

Serra Lleti et al. report a new software (CLEMSite) for fully automated FIB-SEM imaging based on locations identified beforehand in LM. The authors have implemented routines for automatically identifying common reference patterns and an automated FIB-SEM quality control. This allows autonomous data acquisition of multiple locations distributed over the entire sample dish. CLEMSite has been developed as a powerful tool for fast and highly efficient screening of morphological variations.

****Major comments:****

The performance of CLEMSite has been demonstrated by the authors with two typical biological example applications. The stated performance parameters such as correlation precision and reproducibility are highly convincing and supported by the presented data. The authors give detailed information on their workflow and how on to use CLEMSite, which should allow other researchers to implement this for their own applications. The only comment I have in this regard, and I might have overlooked it, but how will CLEMSite be made available to the scientific community?

Reply 1.1

We would like to warmly thank Reviewer #1 for their very supportive feedback. It is important to us to share our work with the community. Our prime intention is to offer CLEMSite as a proof of concept that has been demonstrated on a specific instrument, thus linked to a company (Zeiss). Because we are convinced this code can be adapted to other APIs provided by other vendors, we made it fully available via a Github repository (<https://github.com/josemiserra/CLEMSite>).

Full Revision

To make this more visible in the manuscript, we have modified this sentence to the first paragraph of the Results section:

“To control the FIB-SEM microscope, CLEMSite-EM interfaces commercial software (SmartSEM and ZEISS Atlas 5 from Carl Zeiss Microscopy GmbH) via a specific Application programming interface (API) provided by Zeiss. CLEMSite code is openly accessible and free to download from a Github repository (<https://github.com/josemiserra/CLEMSite>).”

****Minor comments:****

The author mention that decreasing the z-resolution to 200 nm steps was critical to achieve high throughput. For applications that require higher resolution: is the only disadvantage a longer data acquisition time or are there also other limitations?

Reply 1.2:

Reviewer #1 is right, we have designed CLEMSite as a screening tool, where we emphasize the number of cells versus the resolution at which each cell is acquired. By acquiring images every 200 nm, we are gaining speed, but also stability. We have indeed noticed that below 50 nm, on occasions the beginning of the acquisition is not stable enough (the milling has to hit the front of the cross-section at view precisely), and it requires manual intervention to retract/advance the milling. In addition, to gain time in our current workflow, we have opted to not cover the region of interest with a platinum protective layer, which has no consequences when imaging at larger z steps because the overall time spent on one cell is very short. At higher z resolution regimes though, a non-protected block surface is inevitably damaged during the successive numerous mill & view cycles. We have added one sentence in the Methods section to make this point clearer.

“Note that to gain time in the preparation process for a run, we have not covered the ROI with a platinum protective layer and alternatively we increased the thickness of the gold coating of the full sample. In such cases, only low z-resolution acquisition is possible, as acquiring at a higher resolution would require sputtering of the sample surface.”

Finally, we may argue that if an experiment requires high-resolution acquisition, the time overhead spent to switch from one cell to the next (a few minutes) is not significant anymore relative to the time spent to acquire one cell (from several days to weeks). In such cases, automation for multi-site acquisitions would lose its relevance.

I would assume that locating the finer structural details in a much larger data set might also introduce additional challenges in the data analysis pipeline.

Reply 1.3:

We fully agree with Reviewer #1. In this proof of concept study though, we are not addressing the image analysis part but assess ultrastructural phenotypes manually using established stereology protocols. At the image resolution that we are using, our analysis is restricted to features such as volumes, surfaces, number of rather large organelles. Finer details, such as microtubules or fine contact sites between organelles would require a higher resolution, and indeed very likely other means to extract the morphometric data. State-of-the-art image analysis of isotropic FIB-SEM datasets is based on computer vision/machine learning. With such tools, the analysis of fine details is indeed accessible with very high accuracy, but at the cost of the throughput, at least for now as already mentioned in the Discussion section of the paper.

In Table 1 in the supplements, the units are missing for the targeting positions. On page 4, 4th line from the bottom, there is a typo in "reaaching a global targeting...".

Reply 1.4

We thank Reviewer#1 for their thorough inspection of the paper. We have corrected it accordingly.

Reviewer #1 (Significance (Required)):

With CLEMsite, the authors present a powerful new software tool for the FIB-SEM imaging community. The high level of automation allows high throughput data acquisition with minimal user interaction. To my knowledge, this is the first software that fully automatically recognises reference features and is able to run fully autonomously after points of interest have been selected in FM. This high throughput screening tool for FIB-SEM imaging would make a substantial technical contribution to the field of cellular imaging. My own expertise lies in the field of technical developments for CLEM and super-resolution FM. I am not able to judge the biological content of the manuscript.

We would like to thank Reviewer #1 for their constructive and encouraging feedback.

Reviewer #2 (Evidence, reproducibility and clarity (Required)):

Review on "CLEMsite, a software for automated phenotypic screens using light microscopy and FIB-SEM" by Serra Lleti et al. The manuscript describes a toolset to correlate LM data with automated FIB-SEM of selected regions of interest. This allows 3D correlative microscopy of multiple adherent cells from a single resin block. This allows much needed high throughput in CLEM analysis to become quantitative. Two applications on Golgi apparatus morphology are shown.

****Major questions:****

-The software has been developed in collaboration between Zeiss/ Fibics in collaboration with academic groups and will only function on Zeiss SEMs that have the proper software. Thus, if I understand correct, it will not be of generic use and a more appropriate title would be 'CLEMsite, a software for automated phenotypic screens using light microscopy and Zeiss FIB-SEM'

Reply 2.1.

Reviewer #2 is right about the fact that our work was done on a Zeiss microscope and in CLEMsite's current version, it would only work with that model, including firmware and software. As already phrased in the manuscript, we would like to stress our work is a proof-of-concept. For example, we wrote in the introduction that CLEMsite is a "software prototype". We've also made clearer the links to Zeiss in the first paragraph of the Results (see also answer to reviewer 1.1) CLEMsite is by no means designed to become an integrated part of current or future Zeiss microscopes. On the contrary, we have designed the software as an independent unit. All the parts of the software that are sending commands to the Zeiss API are indeed customized to that brand, but other functions are stand-alone units. In particular, the correlation strategy is

independent of the microscope type and can be used generically. Similarly, the principles that we developed for finding the FIB-SEM coincidence point, or for selecting features-rich regions to perform the AFAS function would be valid whichever microscope model would be used. For these reasons, we would prefer to avoid mentioning Zeiss already in the title of the manuscript.

- How is the described approach using FIB-SEM advantageous compared to methods like Serial Block-face EM (SBEM) and array tomography using serial section where larger fields of multiple cells can be imaged? Especially because the axial resolution was set to 200 nm and discussed as essential for the throughput speed.

Reply 2.2.

This is a very important point that we tried to bring across in the introduction of the manuscript. Other volume EM methods, such as SBEM and AT, like conventional TEM, require an ultramicrotome to produce thin sections (AT and TEM) or to remove the top layers from the resin block (SBEM). This inevitably requires trimming large specimens in order to accommodate the dimensions of the diamond knife used in the associated microtomes. FIB-SEMs does not have such limitations and selected volumes can be imaged from samples of any size, providing they fit in the chamber of the microscope. In our case, we were screening cells growing on a 1 cm² surface area, which is already beyond what standard diamond knives can process. We would even argue that larger surfaces are at CLEMsite reach, but we have not tested this.

- Is the data FAIR available?

Reply 2.3

It is one of EMBL's ambitions to make all data as FAIR as possible. For this study, we saved all the raw images and their corresponding embedded metadata as they came from the original software (ATLAS 5, Fibics for the SEM images and LAS X, Leica microsystems for the confocal images). The images published in this manuscript will be deposited on the EMPIAR data repository upon acceptance. The raw data and unpublished data, due to their size, will be fully available upon request to the authors. Additionally, their data is specifically generated for the correlation workflow, which is stored together with the image information as separated files. To store the information of logs we used text files, for intercommunication between processes, JSON, and XML to store coordinates in a readable format. As far as we know, there is no standard FAIR protocol yet that describes CLEM workflows in microscopy. We made our best possible efforts to archive our data in an understandable folder architecture, with detailed information on how to navigate through it, such that we are confident that our data could be mined by others in the future, thus reaching the goals of the FAIR charter.

- How is CLEMsite available? Is the code public or for sale?

Reply 2.4

It is important to us that our proof-of-concept can be used or adapted by others in the future. For this reason, we are sharing the full code that was developed for CLEMsite - See Reply 1.1 for further details.

****Other comments:****

- Can you comment on the flexibility of this method? It is described as a flexible method, but only HeLa cells (quite flat cells) and Golgi apparatus targeting was used. What about different cell types and what about targets with a less obvious EM morphology?

Reply 2.5

It is correct that we have demonstrated our workflow only on HeLa cells which present a more or less homogeneous topology. Yet our workflow is flexible when it comes to the dimensions of the region of interest and the acquisition field of view, and can accommodate a wide range of cell shapes, as long as they adhere to a culture substrate. Dimensions of ROI and FOV can be adapted in the CLEMsite interface as described in Supplementary Figure 4. Following reviewer 2 question, we realize that this feature may not appear clearly and we have modified the corresponding section of the Result:

“The dimensions of the image stack, as well as the z resolution are set when initializing the run, via the CLEMsite interface (Supplementary Figure 4). Whilst every cell of one run can be acquired with the same recipe (as defined in *ZEISS Atlas 5*: sample preparation, total volume to be acquired, slice thickness and FIB currents applied at each step), CLEMsite-EM also offers individual definition of recipes, allowing a per cell adaptation of the shape or volume (Supplementary Fig. 4a).”

Changing the ROI size would thus accommodate the surface occupancy of a cell (in the plane parallel to the culture substrate) and changing the FOV would accommodate the cell's height.

The morphology of the cell as it appears in the EM (SESI) does not alter the targeting strategy, since we are solely relying on the correlation, which means that the position of the target cell is extracted from the light microscopy images and the coordinate system provided by the gridded coverslip. Even if the cells were invisible at the surface of the resin block when inspected in the SEM, CLEMsite would still navigate to the proper region and create an image stack by FIB-SEM imaging.

- For EM acquisition ZEISS smartSEM with ATLAS was used. LM was recorded with a microscope from a different vendor. Can the software be used regardless of microscope type?

Reply 2.6

Yes, the correlation is based on collecting the stage coordinates from the light microscope, and on analyzing the images from the various magnifications and channels. This information can be obtained by most microscope types, but it might involve minor adaptations regarding the specific brand of a microscope (e.g. changes in the coordinate system of the stage used or the naming of the channels).

- Create less variation in the size of scale bars.

Reply 2.7

We have modified all figures to take this comment into account and thank Reviewer #2 for a good suggestion.

- M&M: High-resolution light microscopy: Why call this 'high resolution'?

Reply 2.8

Full Revision

We used this term to differentiate, in the feedback microscopy setup, the first stage where images are acquired at low magnification from the images acquired at high magnification. We agree that the term is misleading, so we decided to update the manuscript and change the term high resolution by higher magnification (the second stage in feedback microscopy).

Specs given seem randomly chosen: For example objective magnification yes, NA not; excitation wavelength yes, emission not.

Reply 2.9

We thank Reviewer #2 for spotting these missing details. We have edited the method section to add the NA and the emission wavelengths.

Reviewer #2 (Significance (Required)): See above: This depends on the availability of code, as well as the usability in FIB-SEM that is not based on Zeiss.

Reply 2.10

We hope our answers have addressed these concerns. When the code is indeed fully available, we can not at this stage presume of the transferability of CLEMsite to microscope from other manufacturers. Yet we would like to stress once more that our main aim is to demonstrate a proof of concept.

Reviewer #3 (Evidence, reproducibility and clarity (Required)):

****Summary:****

Schwab and coworkers present an automation software for correlative light and electron microscopy (CLEM) to acquire high-resolution SEM volume datasets at room temperature. This automation enables large-scale data collection and morphometric analysis of cell phenotypes. The paper is overall well written, but often assumes a lot of prior knowledge of the workflow, which might not be present in a general audience or for newcomers to the technique. This is also seen in the insufficient labeling and explanation of the figures. They seem a bit like presentation slides, which could be well understood with the help of the presenter/narrator, but alone lack a lot of information (see more specific comments below).

****Major Comments (in no particular order):****

- Final accuracy of $\sim 5 \mu\text{m}$... is this really sufficient? Given that the size of many mammalian cells is $\sim 10\text{-}15 \mu\text{m}$, this is still a HUGE error. Of course, there is a tradeoff between throughput and accuracy, the area covered and speed. Nonetheless, this means a serious limitation in terms of the kind of targets / biological questions that can be addressed with this technique! (Especially in the context of "rare events") This should be discussed in more detail.

Reply 3.1

We thank reviewer #3 for their constructive criticism of the work. Indeed, our final accuracy is $5 \mu\text{m}$ at best, which may at first glance appear as a disappointing value. This accuracy is the consequence of a couple of strategic decisions that we have made in designing the workflow, which will be further explained below. We have chosen to constantly opt for a large field of view that would be larger than the average cell size, thus mitigating the potential $5 \mu\text{m}$ offset in

targeting. In our hands, this yielded satisfying results, yet we agree that a higher targeting precision would allow narrower fields of view and potentially an even increased throughput.

Our correlation strategy fully relies on the coordinate system built from the gridded pattern embossed at the surface of the culture dishes. The precision of CLEM*Site* automated targeting thus relies on i) its ability to properly detect the grid edges, both at the LM and at the ME, and ii) on the mesh size of the grid. To ensure a wide range of applications, we decided to design CLEM*Site* on commercial culture dishes, of which the MatTek gridded culture dishes appeared the most convenient, for each grid square presented a unique alphanumeric ID together with a relatively large and flat surface area to accommodate a large number of cells away from the grid pattern. Whilst such dishes showed a topology that satisfied our first criteria, the grid spacing was 600 μm . A smaller mesh size would have undoubtedly resulted in higher precision in the targeting but at the expense of losing free areas. Other commercial dishes with denser meshes unfortunately would not have ID engraved directly inside every square or we experienced difficulties in reproducibility during the sample preparation process to detach the glass from the resin block.

We also have excluded the option to design our own grids, which would have created another dependency for potential users from other laboratories.

Another possibility for targeting would be to register the fluorescence maps to the shapes of the cell as visible in the resin block. Adherent cells can be detected in the SEM if high energies are used to scan the surface of the blocks, and also if the block is not coated with a too thick layer of gold. In our experience, switching between voltages for acquiring such overviews and low voltages for acquiring FIB-SEM stacks is another source of imprecision and doesn't improve the targeting in very confluent areas. Another interesting idea, as shown in Hoffman et al 2020, would be to scan the embedded samples by X-ray prior the FIB-SEM targeting, but not only this would imply that high-end X-ray machines would be available for such tasks, but would still require landmarks to register the X-ray maps to the SEM overviews. This would potentially yield a higher accuracy, but we have opted for the gridded substrates, judged more accessible to a large number of laboratories.

We tried to explain such a choice in the discussion, by adding this sentence in the description :

“Detection of local landmarks imprinted in the culture substrate enables automated correlation and targeting with a 5 μm accuracy. We estimate that this number could still be improved by customizing a gridded substrate with a smaller mesh size, as landmarks would be much closer to the targets. The detection algorithm we developed could be extrapolated to other customized dishes or commercial substrates for cell culture in SEM samples⁴¹. An advantage of using local landmarks for the correlation is that they mitigate the impact of sample surface defects or optical aberration across long distances. Alternatively, targeting individual cells with a FIB-SEM has been achieved by mapping the resin embedded cells with microscopic X-ray computed tomography⁴⁴. We speculate that such tools could be an alternative to a gridded substrate, yet cannot predict its adaptability to large resin blocks such as the ones we used in this study. “

• Given that the whole point of the paper is "large scale automation", I would have preferred a few more examples/higher n-count. A comment on which type of targets the authors envision/have validated would be nice (also in the context of the limitation in accuracy).

Reply 3.2

To our best knowledge, no one has ever imaged multiple cells automatically. So even 5 in a row is a high number.

In addition to this, we added an extra paragraph in the discussion.

“We believe that other research questions could benefit from this type of screening. As an example, the 2021 Human Protein Atlas Image Classification competition⁶¹ managed to classify multiple organelles of individual cells in fluorescence microscopy. Such machine learning models could be used to find rare events or particularly interesting phenotypes. In another example, in host-pathogen interactions, early infected cells might start to display a recognizable phenotype in a small subpopulation of cells⁶². In both cases, those marked cells could be used to establish a FIB-SEM screening to discover new morphological differences at the micrometer level.

To expand the applicability of these screenings beyond the proof-of-concept here presented, we propose two directions of improvement. First, by acquiring smaller enclosed volumes with isotropic resolution, we could target area-delimited organelles, like centrioles²¹. In this case, the full cell volume is neglected in favor of a small portion of it, but with higher z resolution. At the software level, that would require improving targeting accuracy by using smaller grids and extending the maps to 3D coordinates. 3D registration against a light microscopy Z-stack would considerably help to constrain the field of view during acquisition, thus reducing the imaging time and keeping the field of view position during tracking. At the instrument level, this would require, first, stabilizing the ion beam before the critical region is acquired, to compensate for the change between high currents for milling and fine currents for sectioning. Second, to make sure that the fine current beam hits exactly the front face of the milled cross-section, and then prevent milling artifacts. Finally, the second direction is to increase the number of samples acquired per session. That would imply ion beams that automatically reheat the Gallium source when it is exhausted (like proposed in Xu et al. ⁶⁰), with faster algorithms for autofocus and autostigmatism in SEM.”

- It should be mentioned somewhere that "commercial dishes or coverslips" contain an imprinted grid pattern with numbers and letters to locate specific squares. [Again: probably clear to "aficionados" of the technique but totally unclear to newcomers/outsideers]

Reply 3.3

We have added an explanation of the layout of the coordinate system in the part of the correlation strategy and the methods section “gridded dish with numbers and letters” to explain the correlation and targeting strategy better.

- "It is important to keep the initial number high in order to compensate for the loss of targets" - what % of targets is lost exactly in the final step (FIB-SEM imaging)? The 10 cells out of 35 (29%!) that were not "of sufficient quality for further downstream analysis", were they lost/discarded because of problems in the automation (e.g. autofocus/tracking failure) or for other reasons (e.g. preservation of the cells during fixation/embedding)?

Reply 3.4 In the main text we decide to explain the process of filtering better. We have added a supplementary figure showing different causes for problems of coordinate system detection due to scratches, cracks, or dirt. None of the cells in the study were discarded due to bad preservation, but the system being a proof of concept, we dealt with multiple difficulties that forced us to filter the acquired stacks for getting the ones showing the best quality. We also added supplementary

material (Sup. Tables 3 and 4 with explanation) about the possible causes of cell losses during different experiments.

- "One essential paradigm shift for increasing the acquisition throughput is the decision to decrease the resolution in the z-dimension, thus prioritizing the speed of acquisition and ultimately the total number of cells acquired in one run.". Surely, reducing z-resolution is an obvious way to speed up acquisition times. But this is not tied to the use of this software and obviously comes at a price ... this has been discussed before and is nothing novel. Hence "paradigm shift" might be a bit too strong. I however fully agree with CLEMSite's potential as a screening tool. Could a "high resolution" (isotropic) mode not be implemented, too? [then it would be up to the user to decide what to prioritize - throughput or resolution]

Reply 3.5

We have replaced the word "paradigm shift" with "original strategy". It is indeed up to the user to decide if higher z resolution or higher speed should be achieved by setting up different recipes. Additionally, we direct the reviewer to read Reply 1.2.

- There is no mentioning of why this specific hardware was used. Are there any limitations that currently restrict the approach to Zeiss machines? Any plans supporting other vendors? Of course, there are always certain benefits with certain instruments. Or just simply no others were available... A comment on which part was performed by/at Zeiss and which in the labs would be useful to understand specific contributions. (Since a conflict of interest statement seems missing).

Reply 3.6

The original plan was to set up a proof-of-principle study developing a program that is fully open source. We created an interface, which could plug any control via proprietary API, by simply adapting commands from the API to our interface. The idea is very similar to what is done in light microscopy open-source controllers, like the Micropilot software (<https://www.ncbi.nlm.nih.gov/pmc/articles/PMC3086017/>).

That interface would be the place where to modify the software and add the external API and requires only that such API can be used with a .NET framework in C#. The programmer would have to modify only the following file:

<https://github.com/josemiserra/CLEMSite/blob/master/CLEMSiteServer/TestApp/AtlasCom.cs>.

We expect that FIB-SEMs are very similar across companies, at least in basic functionality (get images, get positions, mill execution for trench digging with a recipe file, ...). We thus believe our software could be adapted to other vendors. As an example, we used the Fibics API, but we could also program the same functionality with the Zeiss API to achieve the same goal.

Zeiss's contribution to the project was i) providing a system during the initial phase of the project, and allocating time with programmers from FIBICS to help to provide the control API to be used by CLEMSite.

All experiments were performed at the European Molecular Biology Laboratory or Max-Planck Institute of Experimental Medicine.

****Figures:****

Figures should be improved. They often contain too little information to understand the concepts/results discussed and there's lots of white space. The legends should be improved accordingly. In general, a more concise and structured figure design could go a long way of improving the quality of the manuscript. Please find a few suggestions (for the main figures) below (but the same should be applied to the supporting figures):

Reply 3.7

We thank the reviewer for the suggestions on the figures in general. We have revisited all the figures and made corresponding changes as highlighted below.

Fig. 1: While I believe it is clear to me what each scheme is supposed to represent, someone less immersed in this topic (or just entering the field) may have problems navigating the figure. For example: what are all the different letters and numbers? What's the blue box with the trapezoid ("EM targets" - it may become clear later, but here it is not), what are the blue and the red arrowhead, respectively (I suppose EM and focused ion beam?). This should be improved and labeled accordingly.

We have addressed the queries by explaining the figure more explicitly in the legend (e.g. blue box, blue and red arrowhead). We have added i, ii, iii to separate b into subsections and adjusted the text accordingly.

Fig. 2: Again, a lot of annotation is missing. E.g. what is the 3rd insert in b exactly (edge-detection? After CNN identification?)? For most of the figure, yellow squares are used to indicate the zoom-in region, why not for b) 1st row? With the "zoo" of scale bars, wouldn't it make sense to either always show the same bar (e.g. 200 μm), or scale the images so things become more comparable? In this regard: a) 2nd column and e) 1st column represent the same FOV. Why are they shown with different magnification/cropping?

Reply 3.8

The scale bars have been homogenized when possible, in the case of b the image was zoomed to match a (first image in both cases). A yellow box was added for the zoom in b as suggested. We added descriptions in the figure legend.

Fig. 3: a) The procedure is well described in the legend, but no motivation is given in the text, why this is necessary. c) There's some floating density in the white space. Is this due to thresholding? Already explained in figure legend.

Reply 3.9

We have adapted the text in the main manuscript to explain better that the coincidence point is normally found manually and for a routine with as little as possible intervention by an operator this had to be automated. We have also explained figure 3c more in the figure legend.

"The following steps, usually performed by a trained human operator, are triggered autonomously: localization of the coincidence point, needed to bring the FIB and SEM beams to point at the same position (Fig. 3a); milling of the trench to expose the imaging surface, detection of the trench to ensure a well-positioned imaging field of view (FOV) (Fig. 3b); automated detection of image features in the imaged surface needed to find an optimal location for the initial autofocus and autostigmation (AFAS) (Fig. 3c); and finally the stack acquisition (Fig. 3d)."

Fig. 4) Again, description/labeling of the figure is poor. E.g. what are the red outlines present in c) row 1-3 (but missing in row 4; why?)? [presumably these are the siRNA spots?] Is there any reason this figure could not be further subdivided into d), e), f) etc? As it stands, a lot of additional descriptors ("second from the left", "two images on the right") are necessary while a simple call to a), b), c) would be much easier...

Reply 3.10

We have more precisely described the siRNA spots in the legend more explicitly and have added headings to divide part c into a grid rather than adding letters/numbers to subdivide to make the figure more clear.

Fig. 5) Additional labeling (a,b,c...) could be helpful here, too. While intuitively I would assume that blue = DAPI and green = GFP, these things should be labeled or described in the legend. Especially in the 3D rendering it is quite unclear what is being portrayed. Is this an overlay of a FIB/SEM segmentation with the confocal 3D-data?

Reply 3.11

We have added headings to subdivide the images in b and explanations in the legends to explain the color-dye relation (blue is DAPI).

****Minor:****

The first element in the filtered list can thus be stored for the subsequent application of autofocus and autostigmation procedures (AFAS) (Supplementary Fig. 3c). [technically this has been defined before]

Reply 3.12

All typos and grammar-related issues have been addressed in the following ways:

A transformation is computed to register together the LM list and EM landmarks list, ...

"A transformation is computed to register the positions from the LM and the EM landmarks list, ..."

"The FOV is changed from a 305 \$\mu\text{m}\$ by 305 \$\mu\text{m}\$, used for the detection of the trench, to a 36.4 \$\mu\text{m}\$ by 36.4 \$\mu\text{m}\$ in the exposed cross-section and an image of that cross-section is taken for analysis (Fig. 3c)."

The sample is positioned at the target coordinates of the first cell, and the Multisite module performs the coincidence point alignment of both the electron and ion beams (Fig. 3a and Supplementary Fig. 2a).

Reviewer #3 (Significance (Required)):

****Significance:****

It is clear that the kind of automation outlined here is necessary to elevate correlative SEM volume imaging to a "high-throughput" technique, which could become valuable for many biological questions. CLEMsite offers a valid technical solution and appears to be a solid implementation of

Full Revision

the traditional/manual workflow. However, its presentation needs to be improved before we can support publication.

Reply 3.13

We have worked on different aspects of the presentation, rearranged the figures, and extended figure legends and hope this meets the reviewer's expectations.

November 10, 2022

RE: JCB Manuscript #202209127T

Dr. Yannick Schwab
European Molecular Biology Laboratory
Meyerhofstrasse 1
Heidelberg 69119
Germany

Dear Dr. Schwab,

Thank you for submitting your revised manuscript entitled "CLEMSite, a software for automated phenotypic screens using light microscopy and FIB-SEM." The manuscript has now been re-reviewed by two of the original reviewers from Review Commons. We would be happy to publish your paper in JCB pending final revisions necessary to meet our formatting guidelines (see details below).

A. MANUSCRIPT ORGANIZATION AND FORMATTING:

- 1) Text limits: Character count for Tools is < 40,000, not including spaces. Count includes title page, abstract, introduction, results, discussion, and acknowledgments. Count does not include materials and methods, figure legends, references, tables, or supplemental legends.
- 2) Figure formatting: Tools may have up to 10 main text figures. Scale bars must be present on all microscopy images, including inset magnifications.
- 3) Statistical analysis: Error bars on graphic representations of numerical data must be clearly described in the figure legend. The number of independent data points (n) represented in a graph must be indicated in the legend. Statistical methods should be explained in full in the materials and methods. For figures presenting pooled data the statistical measure should be defined in the figure legends. Please also be sure to indicate the statistical tests used in each of your experiments (both in the figure legend itself and in a separate methods section) as well as the parameters of the test (for example, if you ran a t-test, please indicate if it was one- or two-sided, etc.). Also, if you used parametric tests, please indicate if the data distribution was tested for normality (and if so, how). If not, you must state something to the effect that "Data distribution was assumed to be normal but this was not formally tested."
- 4) Materials and methods: Should be comprehensive and not simply reference a previous publication for details on how an experiment was performed. Please provide full descriptions (at least in brief) in the text for readers who may not have access to referenced manuscripts. The text should not refer to methods "...as previously described."
- 5) For all cell lines, vectors, constructs/cDNAs, etc. - all genetic material: please include database / vendor ID (e.g., Addgene, ATCC, etc.) or if unavailable, please briefly describe their basic genetic features, even if described in other published work or gifted to you by other investigators (and provide references where appropriate). Please be sure to provide the sequences for all of your oligos: primers, si/shRNA, RNAi, gRNAs, etc. in the materials and methods.
- 6) Microscope image acquisition: The following information must be provided about the acquisition and processing of images:
 - a. Make and model of microscope
 - b. Type, magnification, and numerical aperture of the objective lenses
 - c. Temperature
 - d. Imaging medium
 - e. Fluorochromes
 - f. Camera make and model
 - g. Acquisition software
 - h. Any software used for image processing subsequent to data acquisition. Please include details and types of operations involved (e.g., type of deconvolution, 3D reconstitutions, surface or volume rendering, gamma adjustments, etc.).
- 7) References: There is no limit to the number of references cited in a manuscript. References should be cited parenthetically in

the text by author and year of publication. Abbreviate the names of journals according to PubMed.

8) Supplemental materials: There are strict limits on the allowable amount of supplemental data. Tools may have up to 5 supplemental figures and 10 videos. Please also note that tables, like figures, should be provided as individual, editable files. A summary of all supplemental material should appear at the end of the Materials and methods section. Please include one brief sentence per item.

9) Video legends: Should describe what is being shown, the cell type or tissue being viewed (including relevant cell treatments, concentration and duration, or transfection), the imaging method (e.g., time-lapse epifluorescence microscopy), what each color represents, how often frames were collected, the frames/second display rate, and the number of any figure that has related video stills or images.

10) eTOC summary: A ~40-50 word summary that describes the context and significance of the findings for a general readership should be included on the title page. The statement should be written in the present tense and refer to the work in the third person. It should begin with "First author name(s) et al..." to match our preferred style.

11) Conflict of interest statement: JCB requires inclusion of a statement in the acknowledgements regarding competing financial interests. If no competing financial interests exist, please include the following statement: "The authors declare no competing financial interests." If competing interests are declared, please follow your statement of these competing interests with the following statement: "The authors declare no further competing financial interests."

12) A separate author contribution section is required following the Acknowledgments in all research manuscripts. All authors should be mentioned and designated by their first and middle initials and full surnames. We encourage use of the CRediT nomenclature (<https://casrai.org/credit/>).

13) ORCID IDs: ORCID IDs are unique identifiers allowing researchers to create a record of their various scholarly contributions in a single place. At resubmission of your final files, please consider providing an ORCID ID for as many contributing authors as possible.

B. FINAL FILES:

Thank you for this interesting contribution, we look forward to publishing your paper in Journal of Cell Biology.

Sincerely,

Jodi Nunnari, PhD
Editor-in-Chief
Journal of Cell Biology

Dan Simon, PhD
Scientific Editor
Journal of Cell Biology

Reviewer #1 (Comments to the Authors (Required)):

The authors have addressed all my previous points in very detailed responses, which are all very reasonable. They have updated the manuscript accordingly. I don't have any further comments and don't see, from my point of view, any issues regarding the publication of their manuscript in JCB.

Reviewer #3 (Comments to the Authors (Required)):

The authors have sufficiently addressed my previous comments.

Figures and descriptions have been improved significantly.

I'm looking forward to seeing CLEMsite ported to other microscope systems! ;-)